# Climate-induced range shifts drive adaptive response via spatio-temporal sieving of alleles

Hirzi Luqman [1,2] ✉, Daniel Wegmann [3,4], Simone Fior [1,5] ✉ & Alex Widmer [1,5] ✉

Quaternary climate fluctuations drove many species to shift their geographic ranges, in turn shaping their genetic structures. Recently, it has been argued that adaptation may have accompanied species range shifts via the "sieving" of genotypes during colonisation and establishment. However, this has not been directly demonstrated, and knowledge remains limited on how different evolutionary forces, which are typically investigated separately, interacted to jointly mediate species responses to past climatic change. Here, through whole-genome re-sequencing of over 1200 individuals of the carnation *Dianthus sylvestris* coupled with integrated population genomic and gene-environment models, we reconstruct the past neutral and adaptive landscape of this species as it was shaped by the Quaternary glacial cycles. We show that adaptive responses emerged concomitantly with the post-glacial range shifts and expansions of this species in the last 20 thousand years. This was due to the heterogenous sieving of adaptive alleles across space and time, as populations expanded out of restrictive glacial refugia into the broader and more heterogeneous range of habitats available in the present-day inter-glacial. Our findings reveal a tightly-linked interplay of migration and adaptation under past climate-induced range shifts, which we show is key to understanding the spatial patterns of adaptive variation we see in species today.

Present-day species have persisted through repeated periods of fluctuating climate, exemplified by the Quaternary ice ages (2.58 Mya – present) that occasioned major shifts in global sea levels, continental ice sheets and consequently in the habitats of species[1,2]. Species responded to these changing conditions by shifting their range, adapting or going locally extinct, generating the distribution ranges and patterns of genetic structure that we see today[1,3,4]. By shaping the distribution of genetic variation potentially relevant for climate-related traits, past climate fluctuations may have played major roles in dictating the adaptive potential of species, that is, how able species are to adapt to subsequent bouts of climate-driven selection[5,6]. While

studies on the biotic impacts of climatic change are rife[7–9], few studies consider these genetic legacies of past climate, and fewer still incorporate both past and present, neutral and adaptive, evolutionary processes in their evaluations. Such an integrative approach, however, may be crucial to understand and predict species' evolutionary responses to changing climate[10].

Previous studies of species response to Quaternary climate fluctuations have focused on range shifts and range expansions (hereafter collectively referred to as range shifts[3]), through reconstructions of past distributions based on fossil and contemporary occurrence records, and through inferences of past demography based on

[1]Institute of Integrative Biology, ETH Zurich, Zurich, Switzerland. [2]McDonald Institute for Archaeological Research, University of Cambridge, Cambridge, UK. [3]Department of Biology, University of Fribourg, Fribourg, Switzerland. [4]Swiss Institute of Bioinformatics, Fribourg, Switzerland. [5]These authors jointly supervised this work: Simone Fior, Alex Widmer. ✉e-mail: hl636@cam.ac.uk; simone.fior@env.ethz.ch; alex.widmer@env.ethz.ch

patterns of neutral genetic variation[1,3,11,12]. This focus was motivated by the long-held assumption that taxa are more likely to migrate and colonise adjacent habitat than evolve a new range of climate tolerances[3]. The rationale, in part, was that if species could effectively adapt to cope with past climatic change, they would have been able to persist in-situ without shifting their geographic distribution; in apparent contrast with the numerous evidence for range shifts across taxa[13,14]. This paradigm has been increasingly challenged in the last two

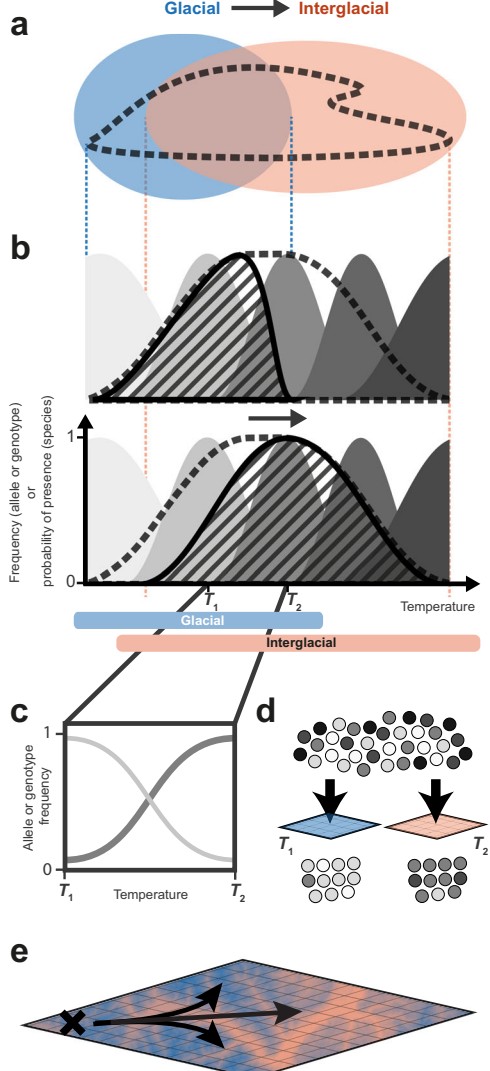

**Fig. 1 | Conceptual mechanism for an adaptive response during a range shift. A** A shift from glacial to interglacial climate leads to a shift in the overlap of a species' fundamental niche (the range of environmental conditions tolerated by the species; dashed contour) with the period's available environmental space (the range of environmental conditions expressed in a certain area and time; blue ellipse – glacial, pink ellipse - interglacial). **B** The species' realised niche (striped curve), reflecting this overlap, hence shifts between glacial (top) and interglacial (bottom) periods, constrained by the species' fundamental niche (dashed outline). Importantly, rather than uniform entities described by a singular niche, species may comprise multiple genotypes, each with their own, potentially different, environmental niche (multiple grey-shaded curves). **C** Climate shifts, e.g., reflecting novel climatic conditions encountered by the species during post-glacial range shifts or expansions, thus act differently on different genotypes; changing their frequencies. **D** This filters or 'sieves' particular genotypes (alleles) from the pool of standing genetic variation depending on the local environment. **E** As the species expands out of a refugia (black cross) and across an environmentally-heterogenous landscape, this drives an adaptive response via the spatio-temporal sieving of climate-associated alleles.

decades[3,5,9,13,15]. Pre-eminently, Davis and Shaw (2001)[3] argued that adaptation concomitant with range shifts may have been central to species responses during Quaternary climate fluctuations. They suggested that adaptation during range shifts can emerge due to selective "sieving"[3,16] of genotypes intolerant to local conditions during colonisation and establishment[3]. In other words, local conditions can act as sieves that sort standing genetic variation unequally across the landscape[3,6,16] during the course of species' range shifts (Fig. 1). This is relevant because adaptation and range shifts, if acting in concert, can lead to vastly different outcomes of species response to climatic change than if either process acts alone[5,15]. Empirical evidence to support this interplay of adaptation and range shifts, however, remains scarce, because of prior-held assumptions and methodological challenges related to the joint reconstruction of these processes[13].

The selective sieving of genotypes across space, and local adaptation in general, imply that populations inhabiting different areas of the species range may carry different genotypes and hence respond to changes in climate differently[5,17,18] (Fig. 1). Accordingly, recent studies have demonstrated that adaptive responses to climatic change can be captured by modelling genotype frequencies to environmental gradients[5,9,17,19], under the premise that each genotype exhibits a range of environmental conditions which it can tolerate. By assuming that contemporary gene-environment associations distributed across space reflect gene-environment associations across time[20,21], these studies suggest that proxies for past or future (i.e., unsampled) genotypes can further be predicted;[5,9,17,20,21] contingent that future or ancestral-like habitats are present and sampled today. While gene-environment models can incorporate adaptation into evaluations of species' response to climatic change, they do not currently integrate other evolutionary processes such as migration and drift, i.e., as shaped by demography[10]. Integration of these demographic processes is, however, increasingly feasible with modern population genetic approaches[10,15], and provides a promising avenue to reconstruct the past neutral and adaptive landscapes of species.

Here, we employ such an integrative approach to elucidate the interplay of adaptation and range shifts in response to Quaternary climatic fluctuations in *Dianthus sylvestris*. This perennial flowering plant native to the Alpine, Apennine and Dinaric mountain ranges of Europe inhabits an environmentally and topographically diverse landscape that was intimately affected by the Quaternary glacial cycles. By quantifying shifts in adaptive genomic composition from inferred ancestral to present-day populations through the novel metric "glacial genomic offset", we show that adaptive responses emerged as a natural consequence of populations' expansion from glacial refugia into the increasingly heterogeneous environmental landscape created with the post-glacial retreat of glaciers. We validate this result by testing glacial genomic offset predictions against observed population genetic signatures sensitive to past selection and demography, demonstrating that we can accurately predict where adaptive diversity is most constrained in contemporary populations. Our results corroborate theory[3,13] that the interplay of adaptation and range shifts has been central in species response to past climatic fluctuations, and highlight the continued role that past fluctuations play in shaping evolutionary responses to climate-driven selection today.

## Results

### Distinct evolutionary lineages separate by geography

We sequenced the genomes of 1261 individuals from 115 populations (5–20 individuals per population) across the contemporary geographic range of *D. sylvestris* at low (mean ca. 2x) sequencing depth (Fig. 2, Supplementary Data 1). Principal component analysis (PCA) and pairwise genetic distances of whole genome sequences separate sampled individuals by geography into Alpine, Apennine and Balkan clusters (Fig. 2A, Supplementary Figs. S1, S2). This was supported by admixture analyses, which

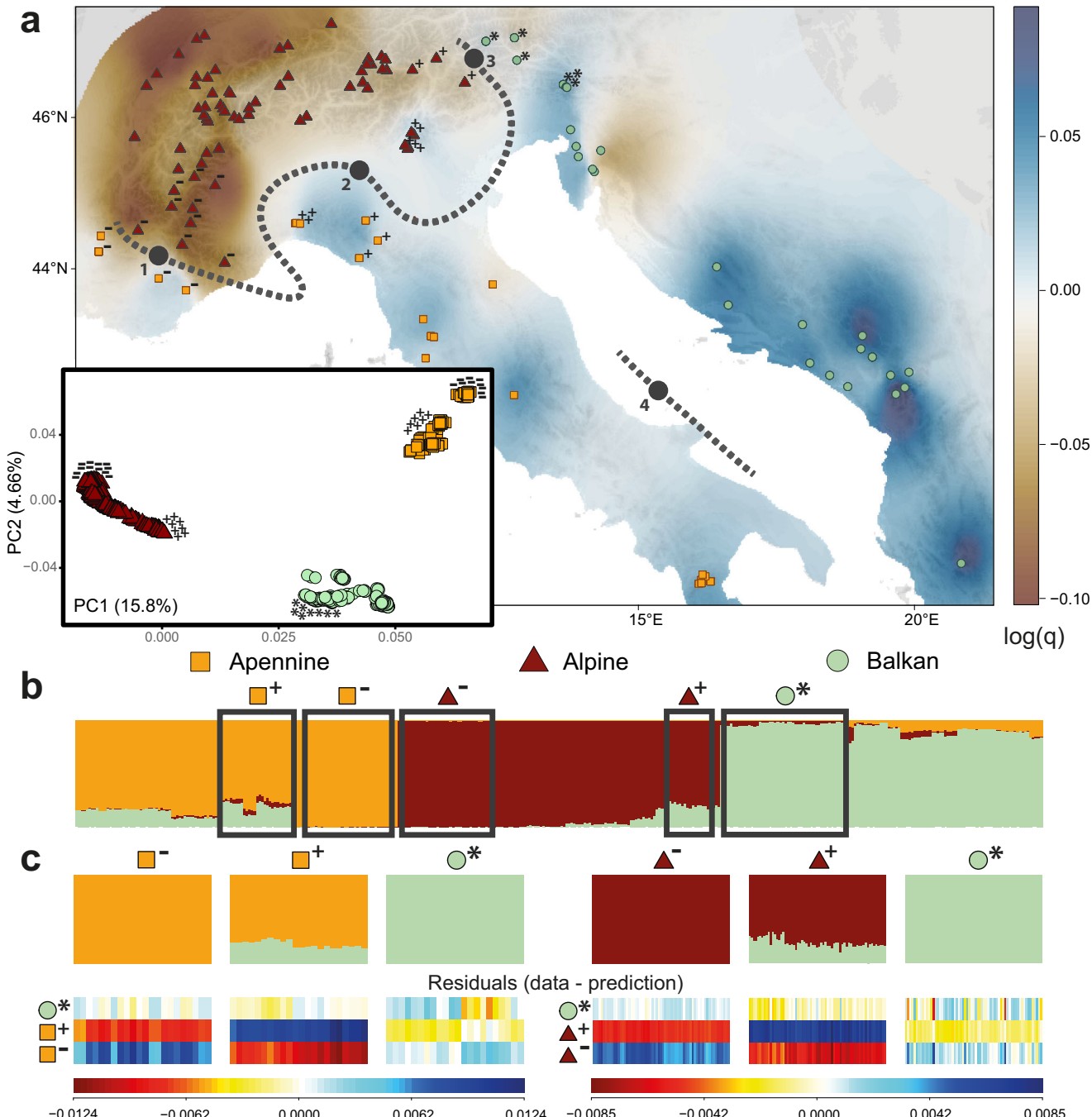

**Fig. 2 | *D. sylvestris* population structure describes genetically and geographically distinct lineages. A** Map of sampled populations. Samples are represented as coloured shapes defined according to genetic cluster, as inferred via principal component analysis (PCA) of whole-genome sequences (PCA results for all sampled individuals in inset; eigenvalues in axes' labels). Landscape colour reflects population effective diversity rates (*q*) calculated via *EEMS*. Thick black dotted lines represent major genetic boundaries as identified via Monmonier's algorithm on populations' $F_{ST}$. Numbers on line are (1) French Prealps-Maritime Alps boundary, (2) Po Plain, (3) Brenner zone, Puster and Gail valleys, and (4) Adriatic Sea. Plus, minus and asterisk symbols denote individuals and populations at contact zones used in chromosome painting analysis. **B** Admixture proportions

of whole-genome sequences at *K* = 3; balanced dataset. Populations are ordered (from left to right): Apennine lineage (south to north-west), Alpine linage (south-west to north-east), and Balkan lineage (north to south). **C** A systematic pattern in residuals (blocks of red or blue; bottom) representing the difference between the observed admixture palettes for select individuals (top) and those reconstructed by chromosome painting supports a scenario of recent bottlenecks in the populations denoted with a minus superscript, over a scenario of between-lineage admixture in the populations denoted with a plus superscript; for the Apennine-Balkan (left) and Alpine-Balkan (right) clines (assessed separately at *K* = 2). Populations used for this analysis correspond to the populations marked with their respective symbol in **A** and surrounded by black borders in **B**.

were performed on a balanced dataset comprising a common, down-sampled size of 125 individuals per geographic region to account for known biases related to uneven sampling across clusters[22,23] (Fig. 2B, Supplementary Fig. S3). To visualise the geographic distribution of the three genetic clusters, we

projected ancestry proportions and principal components in space (Supplementary Figs. S4, S5). We further identified geographic barriers that maximise divergence ($F_{ST}$) between pairs of populations via Monmonier's algorithm[24] (Fig. 2A, Supplementary Figs. S6, S7), and characterised the effective migration surface via

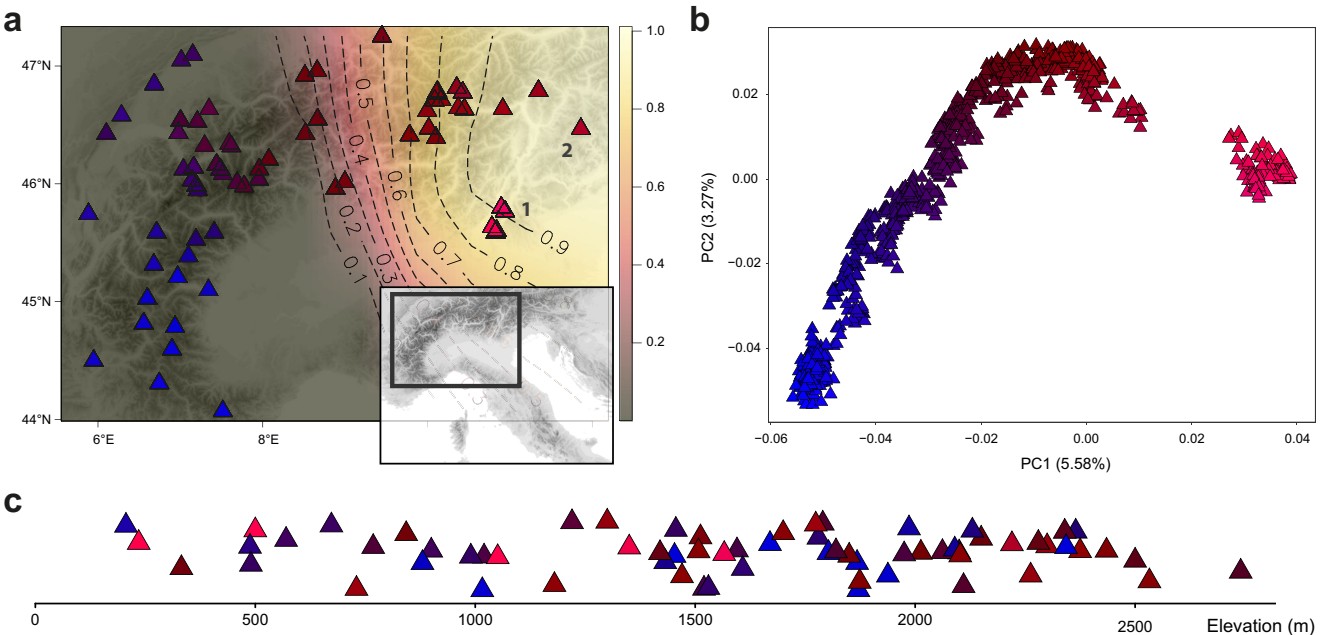

**Fig. 3 | Genes mirror geography in the Alps. A** Geographic sampling of Alpine populations. Black dashed contours and colour gradient reflect the probabilistic inference of the origin of the Alpine expansion, calculated via a time difference of arrival (TDOA) algorithm on the directionality index $\psi$. An eastern origin for the Alpine lineage around the regions of Monte Baldo (1) and the western Dolomites (2) is inferred. **B** Genetic structure of Alpine individuals inferred via principal component analysis (PCA) of whole-genome sequences reflects geography. **C** Elevation of populations in metres above sea level. In all panels, samples are represented as triangles coloured according to their coordinates in the first two principal components of genetic space (as shown in **B**). Population representations in **A** and **C** reflect population means of individuals.

EEMS[25] to describe regions where genetic differentiation is elevated or depressed relative to expectations based on geographic distance (Supplementary Fig. S8). Together, our results point to the Adriatic Sea and Po Plain (Italy) as major biogeographic barriers for *D. sylvestris* (Fig. 2A). In addition, we find that the French Prealps and Maritime Alps form the contact zone between the Apennine and the Alpine clusters in the west, while the Brenner pass and Puster and Gail valleys form the contact zone between the Alpine and Balkan clusters in the north-east (Fig. 2A, Supplementary Figs. S4, S5, S8). The latter is consistent with a well-known bio-geographic boundary for alpine plants[26].

While populations from the separate genetic clusters occur in close proximity in the contact zones (i.e., exhibit parapatric distributions), the PCA of genetic structure suggests deep divergence and consequently a prolonged history of isolation between clusters (Fig. 2A, Supplementary Fig. S2). This is corroborated by demographic inference that estimates the three clusters (lineages) diverged during the Penultimate Glacial-Interglacial Period (PGIP) ca. 200–115 kya (95% confidence interval (CI) for initial Balkan split: 178–217 kya; 95% CI for subsequent Apennine-Alpine split: 114–132 kya), with minimal between-lineage migration in the last ca. 115 kya corresponding to the Last Glacial Period (Supplementary Figs. S9, S10, S11). These results are in contrast to the notable signals of between-lineage admixture observed in the admixture analyses (Fig. 2B). However, inference of population admixture using *STRUCTURE*-like algorithms[27] can arise even in admixture-free demographies because of violations from model assumptions[28]. To tease apart alternative demographic scenarios, we assessed the goodness of fit of an admixture model to the underlying genetic data using patterns of allele sharing inferred by chromosome painting[28]. We observe systematic patterns in the residuals that are more consistent with recent bottlenecks in the western populations of the Alpine lineage and in the north-western populations of the Apennine lineage, than with recent between-lineage admixture scenarios (Fig. 2C). In other words, the increased derived states (uniqueness) of these bottlenecked, peripheral populations conferred

these populations "pure" ancestry assignment under the *STRUCTURE*-derived algorithm, in turn generating false inferences of admixture in the populations that likely founded them (Fig. 2C); a confounding signal that can nevertheless be disentangled using patterns of DNA sharing[28].

Focusing on the Alpine lineage and leveraging off its dense spatial population sampling, we find evidence that the inferred bottlenecks in the Alps reflect a spatial expansion, originating from the east. A cline of genetic diversity reflecting sequential founder events and characteristic of an expansion signal[29,30] is observed along the Alpine arch from east to west (Fig. 2A). This is corroborated by a positive correlation of the directionality index $\psi$[31] with population pairwise distance (Supplementary Fig. S12). Using a time difference of arrival (TDOA) algorithm on $\psi$, we inferred the most probable geographic origin of the expansion to be around Monte Baldo and the western Dolomites (Fig. 3A). We further quantified the strength of this expansion at 1% founder effect generated per 139 km. This spatial expansion left a characteristic clinal genetic structure in the lineage, such that genetic structure currently mirrors geography in the Alps (Fig. 3, Supplementary Fig. S13). This pattern of structure, such that populations are clustered by geography and not ecology, implies that ecotype formation along elevational gradients in the Alps evolved repeatedly in-situ rather than once and subsequently spreading across the range (Fig. 3).

### Distribution models recapitulate genetic data and identify distinct glacial refugia

The inference of three evolutionary lineages each inhabiting separate geographic regions suggests that they may have occupied distinct refugia during the last glacial period. To test this, we modelled the distribution of the pooled species as well as of each lineage separately, based on contemporary occurrence records and present-day climate, and projected these distribution models to the Last Glacial Maximum (LGM)[32]. We applied an unsupervised density-based spatial clustering algorithm on the predicted occurrences, which inferred in both the pooled and lineage-specific cases three discrete refugia: in the Alps,

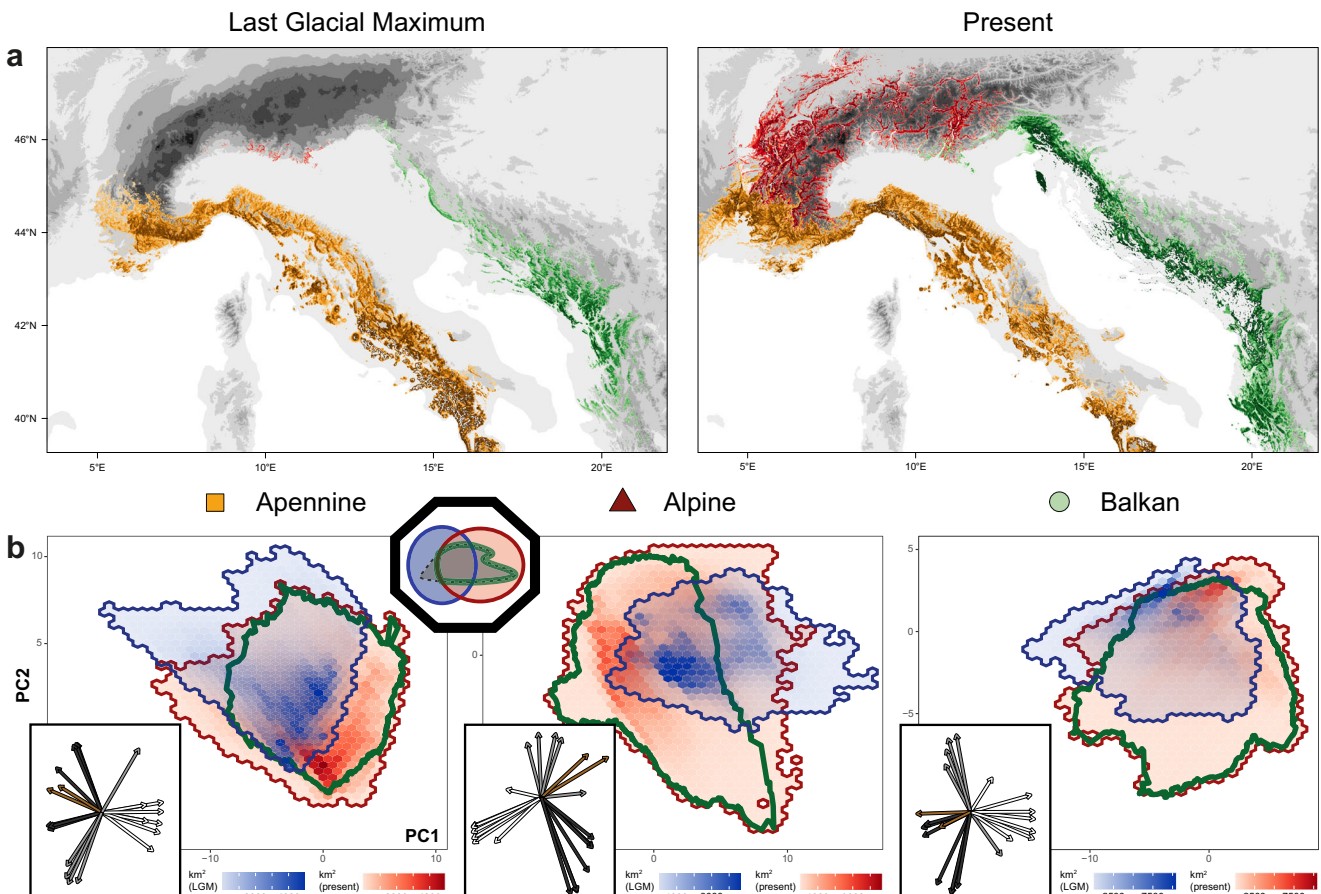

**Fig. 4 | Shifts in environmental space dictate past and present ranges.**
**A** Lineage-specific distributions predicted for the last glacial maximum (LGM) (left; i.e., showing glacial refugia) and present-day (right). The prediction of the present-day distribution recapitulates observed patterns of genetic structure. Topographic elevation is represented in shades of grey and the sea is shown in white. **B** Shifts in environmental space and habitat availability from the LGM (blues) to the present-day (reds), for the Apennine (left), Alpine (middle) and Balkan (right) regions. Each coordinate in this projection of environmental space represents a unique combination of environmental variables. The geographic area (km²) represented by each

coordinate is shown as coloured hex-bins, with increasing red and blue hues indicating greater area exhibiting the indicated combination of environmental variables in the present-day and LGM, respectively. The environmental niche of the lineages, calculated from present-day occurrences, is shown by the green enclosed lines. The octagonal inset recalls the principles introduced in Fig. 1 and provides an aid to interpret these results. The rectangular insets depict the directions and loadings of the environmental variables, with arrows colour-coded as follows: white - temperature variables, grey - climate seasonality variables, black - precipitation variables, and brown - topographic variables.

Apennines and Balkans (Fig. 4A, Supplementary Figs. S14, S15). During the LGM, the distribution range for the Alpine lineage was highly reduced and restricted to the south-eastern peripheries of the Alps (Fig. 4A), consistent with the geographic region independently identified as the source of the expansion from our genetic analyses ($\psi$). In contrast, refugial areas in the Apennines and Balkans were much larger and covered extensive areas across their respective peninsulas. The marked difference in size between predicted refugia for the three lineages is consistent with the lower genetic diversity observed in present-day Alpine populations compared to Apennine and Balkan populations (Fig. 2A).

To reconstruct potential colonisation routes, we projected the lineage-specific distribution models to climate rasters at 100-year time intervals from the LGM to the present-day[32]. We applied a dispersal kernel to limit the rate of dispersal and enforced competitive exclusion among lineages, to address non-abiotic factors that may influence lineage distributions[33–35]. Under this forward-simulation, we tracked the advance of lineage-specific SDMs in time, and generated an expectation of present-day lineage distributions highly congruent with the genetic structure (Figs. 2A, 4A, Supplementary Figs. S4, S5). This congruence suggests that range limits in these lineages are well-determined by climate, competitive exclusion among sister lineages and dispersal limitations. Overall, results from the distribution models

provide support for the three evolutionary lineages each having occupied distinct glacial refugia and each having experienced independent glacial and post-glacial evolutionary histories.

## Shifts in environmental space characterised by warm habitats expanding at the expense of alpine habitats
The range shifts between the LGM and present-day were driven by contemporaneous climate shifts, and reflect the expansion of populations out of the glacial refugia into novel habitats and environmental conditions. To quantify and visualise the shifts in available environmental space from LGM to present, for each region, we projected both present-day and LGM sets of environmental variables to a common, lower-dimensional space and quantified the area occupied at each coordinate in this transformed space. In the Alps, available environmental space was both severely reduced and characterised by colder temperatures during the LGM compared to the present-day (Fig. 4B, Supplementary Fig. S16). As a result, less overlap occurred between the lineage's environmental niche and the LGM environmental space, resulting in the highly reduced distribution range (refugia) during the LGM (Fig. 4A). In contrast, the Apennine and Balkan regions underwent less of a restriction in available environmental space during the LGM, with the latter characterised more by shifts in climate seasonality than in absolute temperature between the two time periods (Fig. 4B). In

addition to the contraction and shifts in environmental space inferred for the three regions, the area occupied by distinct environments was markedly redistributed. Particularly in the Alps, warm habitats expanded in area at the expense of alpine habitats during the postglacial period (Fig. 4B).

### Heterogenous climate shifts drove spatio-temporal sieving of alleles

The heterogenous change in environment between the LGM and present-day combined with range expansion out of glacial refugia imposed a spatially heterogenous and temporally changing selection regime on *D. sylvestris*. To assess whether present-day populations have similar, or altered, compositions of adaptive genetic variants compared to refugial populations, we used gradient forest (GF)[17,36] to model SNP allele frequencies to contemporary environment, correcting for structure. We focus on the Alpine lineage leveraging on our reconstruction of its expansion dynamics, and apply GF on ca. 400,000 exon SNPs that segregate across Alpine populations (Supplementary Fig. S17). To facilitate comparison of climate adaptation across time, we generated distribution models of climate-associated genomic composition for the present-day and LGM by projecting GF models to these two time periods. GF projections infer the ancestral adaptive genotype(s) present in the LGM refugia to be most similar to present-day alpine genotype(s) (dark blue and purple hues; Fig. 5A, Supplementary Fig. S18), as a result of the refugia exhibiting a climate most similar to present-day alpine habitats. The adaptive genotype characteristic of present-day low-elevation valleys (green to bluishgreen hues; Fig. 5A) appears distinct to that of alpine habitats (Fig. 5A, Supplementary Fig. S19). This implies that all contemporary populations, whether lowland or alpine, likely evolved to their current adaptive optima from an alpine-like refugial state. This is supported by the observation that low-elevation populations possess a subset of the adaptive variation present in the high-elevation and refugial populations (Supplementary Fig. S19C).

To predict the evolutionary change populations had undergone, we quantified the differences in genomic composition between time points. We accounted for the effect of expansion and isolation-bydistance (IBD) by performing comparisons between the predicted adaptive genomic composition of each current population and that of its (geographically) closest predicted refugial source during the LGM (Supplementary Fig. S20). This prediction of evolutionary change, which we term "glacial genomic offset", is proposed to reflect both the neutral effect of IBD and expansion (collective drift) from the refugial to the present-day population, in addition to the adaptive response to selection imposed by differences in environment between the presentday location and the refugium. We observe a characteristic pattern of low glacial genomic offset in alpine areas close to the inferred refugia and high values of glacial genomic offset in valleys and regions far away from the glacial refugia (Fig. 5B, Supplementary Fig. S21). This implies that both environmental and geographic distances from refugia underlie the divergence of present-day populations from their ancestral populations in the refugia.

### Population genetics validate glacial genomic offset predictions

Non-random evolution of populations, through processes such as directional selection and demographic expansion-contraction, can be inferred from the genetic data of contemporary populations via perturbations in the site frequency spectrum (SFS) from random expectations[37–39]. To validate our predictions, we correlated the glacial genomic offset with population genetic statistics of present-day populations that capture such SFS biases. We observe a greater excess in high-frequency derived alleles—a characteristic pattern produced by selective sweeps[37,38,40]—with increasing glacial genomic offset (positive correlation with Zeng's $E$[37], and negative correlation with Fay & Wu's $H$;[38] $p < 1 \times 10^{-4}$; Fig. 6A). While demography may also affect

these statistics[37], correlations of glacial genomic offset were stronger with $E$ and $H$ centred around environmentally-associated loci ($r = -0.57$ and 0.63, $R^2 = 0.31$ and 0.38; for $H_{GF}$ and $E_{GF}$, respectively) than with genome-wide estimates ($r = -0.48$ and 0.53, $R^2 = 0.21$ and 0.26; for $H_{GW}$ and $E_{GW}$ respectively).

To further assess the legacy of post-glacial demographic and selective processes on genetic diversity, we compared levels of nucleotide diversity ($\pi$) for contemporary low- and high-elevation populations along the Alpine expansion axis. We find higher levels of diversity around environmentally-associated loci compared to genome-wide ($\pi_{GF} > \pi_{GW}$, Mann–Whitney $U$ test; $p < 1 \times 10^{-15}$; Fig. 6B, C, Supplementary Fig. S22), suggestive of highly-diverged adaptive haplotypes being maintained within populations. Importantly, we observe significantly lower diversity (Mann–Whitney U test; $p < 0.001$) in lowelevation compared to high-elevation populations (controlling for the effect of distance), both for genome-wide diversity ($\pi_{GW}$) and for diversity centred around environmentally-associated loci ($\pi_{GF}$) (Fig. 6B). This can arise due to the colonisation of low-elevation environments from high-elevation populations (founder effect)[41], or alternatively, due to polygenic selection in the low-elevation environment[40], or both. Notably, we observe that this difference in diversity between low- and high-elevation population pairs ($\Delta\pi_{GF}$) increases from east to west along the expansion axis, suggesting that populations simultaneously at the expansion front and environmental margin of the lineage host lowest adaptive diversity (Fig. 6B, C).

## Discussion

The impact of past Quaternary climate shifts on species has typically been investigated through the lens of neutral genetic variation and species distribution models, with adaptive processes either ignored or treated separately[3,13]. The prevailing assumption has been that range shifts, rather than adaptation or an interplay of the two processes, were species' principal response to past Quaternary climate fluctuations[3,13]. While this paradigm has been challenged in recent years, arguments supporting a joint response of adaptation and range shifts have remained largely conceptual[3,13]. Here, by modelling the response of climate-associated alleles in the plant *D. sylvestris* under post-glacial warming (20 kya - present), we show that adaptive processes concomitant with range shifts were central to this species' evolutionary response to past climate shifts. Specifically, increasing regional temperatures and the extensive retreat of glaciers post-LGM led to the emergence of novel warm valley environments in the Alps, generating a heterogenous mosaic of alpine and warm habitats in the path of the colonising species. As *D. sylvestris* populations expanded out of alpine-like glacial refugia into this broader and heterogenous environmental space, warm-associated alleles increased in frequency due to climate-driven selection in the low-elevation habitats. In contrast, populations in alpine habitats retained genotypes closer to those of ancestral populations. This spatially-structured adaptive response supports theory that differential survival imposed by local conditions during migration and establishment selectively sieves out maladapted genotypes[3]. In this study, we capture this spatio-temporal sieving process via gene-environment models and the glacial genomic offset that explicitly integrate the effects of past adaptation, migration and expansion.

Our inference of adaptive shifts hinges on the reconstruction of the evolutionary history and post-glacial expansion dynamics of *D. sylvestris*. Genetic structure and demographic inference reveal that *D. sylvestris* comprises three evolutionary lineages which diverged during the PGIP (ca. 200–115 kya). Post-divergence, these three lineages had relatively independent histories, as inferred from their survival in separate glacial refugia and from a lack of recent admixture between them. The Alpine lineage was particularly affected by the LGM, as evidenced by its highly constrained glacial refugial range relative to present, and the relatively low genetic diversity in contemporary

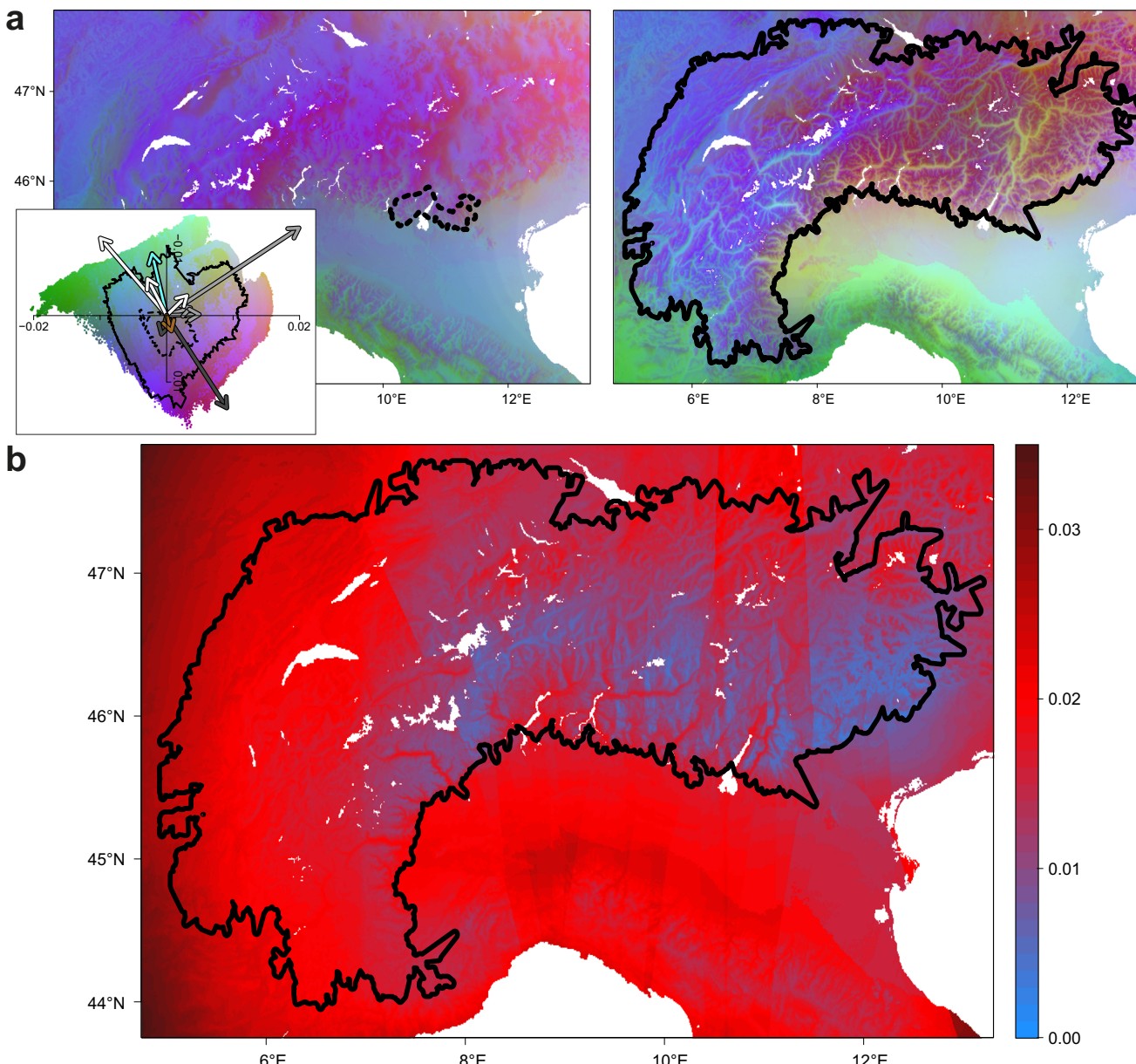

**Fig. 5 | Heterogenous climate shifts drove spatio-temporal sieving of alleles.**
**A** Projections of adaptive genomic composition across geographic space, for the last glacial maximum (LGM) (left) and present-day (right). Areas with similar colours are predicted to harbour populations with similar adaptive genomic composition, while divergent colours indicate populations with divergent adaptive genomic composition. Colours are based on the first three principal components of transformed climate variables (aka eigenvectors describing the composition of genetic variation), where each principal component (PC) is plotted as a separate colour channel in RGB space (PC1 - red, PC2 - green, PC3 - blue). The left inset, applicable to both time projections, shows the loadings (i.e., direction and magnitude) of all environmental variables with respect to the first 2 PCs of transformed climate space, with arrows colour-coded as follows: white - temperature variables, grey - climate seasonality variables, black - precipitation variables, brown - a topographic variable (slope), and aqua - soil pH. The ranges of transformed climate space expressed in the LGM refugium and present-day distribution are shown by the dashed and solid contours of the inset respectively. Dashed and solid contours similarly circumscribe the LGM refugium and present-day distribution of the lineage in geographic space in the main left and right panels, respectively. **B** Prediction of the glacial genomic offset across geographic space. This metric quantifies the evolutionary change in present-day populations from their predicted ancestral state in the LGM refugia, by measuring the shift in adaptive genomic composition between each present-day population and that of its closest LGM refugial ancestor accounting for the effects of expansion and isolation by distance. The present-day distribution of the Alpine lineage is circumscribed by the black line.

populations. Our identification of glacial refugia and expansion routes in the Alps allow for the effects of IBD, expansion and adaptation to be jointly incorporated in our glacial genomic offset, a measure that quantifies shifts in adaptive genomic composition from inferred ancestral to present-day populations. Such an integrative approach addresses previously highlighted shortcomings of gene-environment models[10] and enable us to assess both range shift and adaptation responses simultaneously.

Our retrospective approach provides a unique opportunity to validate evolutionary predictions based on the glacial genomic offset against realised biological outcomes. Population genetic signatures observed across high-low elevation population pairs substantiate glacial genomic offset predictions by showing that populations simultaneously furthest away from glacial refugia and inhabiting environments divergent from the ancestral habitat have lowest levels of adaptive genetic diversity, as a consequence of experiencing the

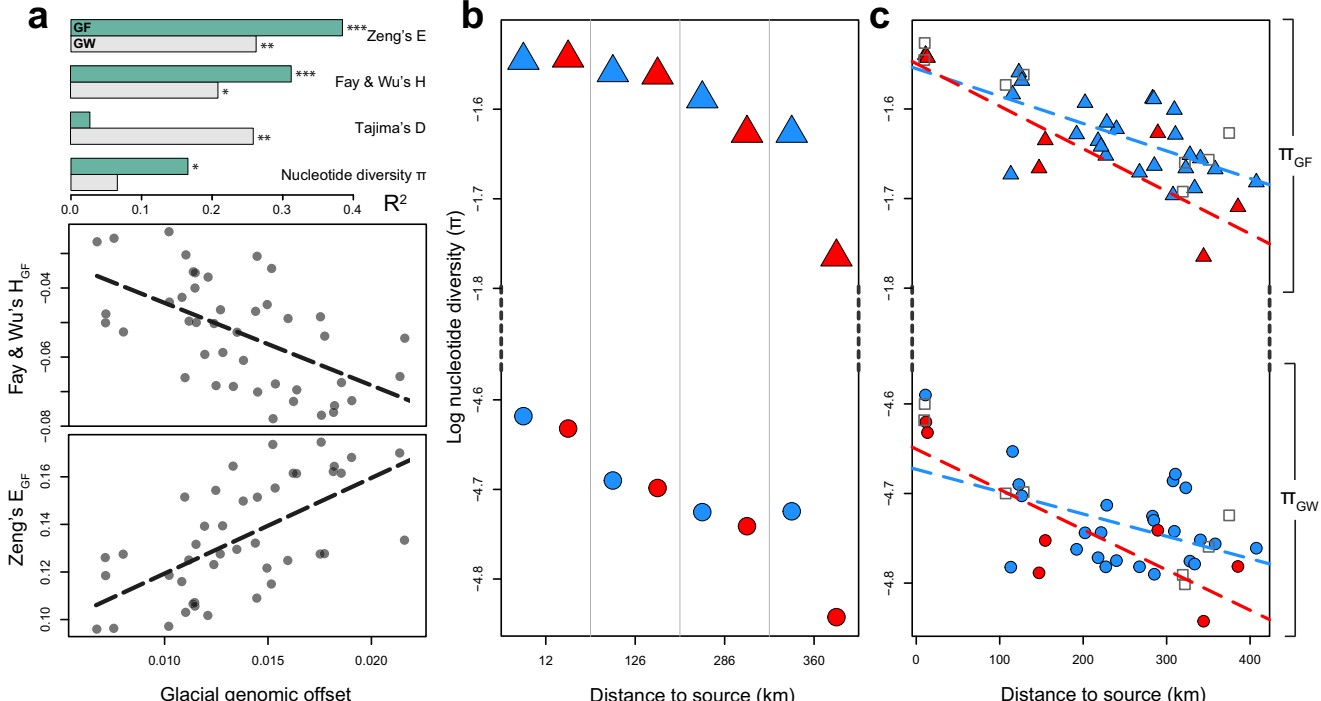

**Fig. 6 | Population genetics validate glacial genomic offset predictions. A** Top panel: Goodness-of-fit (adjusted $R^2$) estimates for the correlation of glacial genomic offset with select population genetic diversity and neutrality statistics (linear regression models; $n = 43$ populations; $p$-value threshold: *0.01, **0.001, ***0.0001; F-test for linear regression). Model goodness-of-fit (bars) are shown for mean per-site statistics calculated genome-wide ($_{GW}$; grey bars) and weighted by the $R^2$ of sites' environmental association ($_{GF}$; green bars). Significant correlations with Fay and Wu's $H_{GF}$ (middle panel) and Zeng's $E_{GF}$ (bottom panel) describe an excess of high-frequency, derived variants - characteristic of selective sweeps - with increasing glacial genomic offset. **B** Plot of nucleotide diversity calculated genome-wide ($\pi_{GW}$; circles) and centred around environmentally-associated loci ($\pi_{GF}$; triangles) for geographically proximate population-pairs with a large (ca. 1000 m) elevation contrast (blue – high elevation; red – low elevation), ordered along the expansion axis. Violin plot of the full $\pi$ distribution can be seen for an example (most-distant) pair in Supplementary Fig. S22. **C** Plot of nucleotide diversity $\pi_{GW}$ and $\pi_{GF}$ for all Alpine populations binned into high (>1500 m; blue), low (<1000 m; red) and intermediate (1000–1500 m; grey squares) elevation classes, ordered along the expansion axis.

most evolutionary change. Correlations of Zeng's $E$ and Fay & Wu's $H$ (population genetic statistics that are sensitive to past selective sweeps) with glacial genomic offset provide complementary validation, and indicate that our model assumptions were justified. Together, this evidence lends credence to our hypothesis that migration and adaptation acted jointly in *D. sylvestris*' response to past climate shifts.

The characteristic imprint left by the concerted mechanism of migration and adaptation on patterns of adaptive genetic diversity has key relevance on how able populations are to respond to future climatic changes. For *D. sylvestris* in the Alps, the described decay of adaptive diversity implies that populations inhabiting the warm, low-elevation environments at the expansion front bear limited potential to adaptively respond to new bouts of climate-driven selection, though predictions can be nuanced. On one hand, low-elevation populations may be relatively well-adapted to tolerate further warming due to past selection in that direction. On the other hand, the consequences of climate change are not exclusive to increase in temperatures, but also novel biotic interactions including competition[42,43] and shifts in other environmental parameters (e.g., precipitation regimes)[44], where the general loss of diversity as observed in low-elevation populations may be detrimental. While the loss of genetic diversity at expansion margins has been well-described[45–48], we show that in heterogenous landscapes, genetic diversity is also determined by the environmental distance between a population's current habitat and that which the population was pre-adapted to (i.e., its ancestral habitat). Thus, our historical perspective can complement future-oriented approaches[9,15,17,19] by providing evolutionary context to observed and predicted patterns of adaptation.

Analysing adaptive variation facilitates our understanding of species response to climatic change because it acts as a lens into the past different than and complementary to that given by stochastic neutral variation, which perceives the demographic past exclusively. Here, we jointly assess patterns of neutral and adaptive genetic variation in an Alpine plant and provide novel evidence for the interplay of migration, adaptation and expansion in the species' Quaternary history. In doing so, we elucidate the manner in which climate acts to shape species' adaptive variation, and highlight species' reliance on the genetic legacies of past climate to respond and adapt to future changes.

## Methods

### Study populations and sequencing strategy
DNA libraries were prepared for 1261 *D. sylvestris* individuals from 115 populations (5–20 individuals per population) under a modified protocol[49] of the Illumina Nextera DNA library preparation kit (Supplementary Methods S1.1, Supplementary Data 1). Individuals were indexed with unique dual-indexes (IDT Illumina Nextera 10nt UDI – 384 set) from Integrated DNA Technologies Co, to avoid index-hopping[50]. Libraries were sequenced (150 bp paired-end sequencing) in four lanes of an Illumina NovaSeq 6000 machine at Novogene Co. This resulted in an average coverage of ca. 2x per individual. Sequenced individuals were trimmed for adapter sequences (*Trimmomatic* version 0.35[51]), mapped (*BWA-MEM* version 0.7.17[52,53]) against a reference assembly[54] (ca. 440 Mb), had duplicates marked and removed (*Picard Toolkit* version 2.0.1; http://broadinstitute.github.io/picard), locally realigned around indels (*GATK* version 3.5[55]), recalibrated for base quality scores (*ATLAS* version 0.9[56]) and had overlapping read pairs clipped (*bamUtil* version 1.0.14[57]) (Supplementary

Methods S1.1). Population genetic analyses were performed on the resultant BAM files via genotype likelihoods (*ANGSD* version 0.933[58] and *ATLAS* versions 0.9–1.0[56]), to accommodate the propagation of uncertainty from the raw sequence data to population genetic inference.

## Population genetic structure and biogeographic barriers

To investigate the genetic structure of our samples (Fig. 2A, Supplementary Fig. S2), we performed principal component analyses (PCA) on all 1261 samples ("full" dataset) via *PCAngsd* version 0.98[59], following conversion of the mapped sequence data to ANGSD genotype likelihoods in Beagle format (Supplementary Methods S1.2). To visualise PCA results in space (Supplementary Fig. S4), individuals' principal components were projected on a map, spatially interpolated (linear interpolation, *akima* R package version 0.6.2[60]) and had the first two principal components represented as green and blue colour channels. Given that uneven sampling can bias the inference of structure in PCA, PCA was also performed on a balanced dataset comprising a common, down-sampled size of 125 individuals per geographic region ("balanced" dataset; Fig. 2B, Supplementary Fig. S3; Supplementary Methods S1.2; Supplementary Data 1). Individual admixture proportions and ancestral allele frequencies were estimated using *PCAngsd* (-admix model) for $K = 2–6$, using the balanced dataset to avoid potential biases related to imbalanced sampling[22,23] and an automatic search for the optimal sparseness regularisation parameter (alpha) soft-capped to 10,000 (Supplementary Methods S1.2). To visualise ancestry proportions in space, population ancestry proportions were spatially interpolated (kriging) via code modified from Ref. [61] (Supplementary Fig. S5).

To test if between-lineage admixture underlies admixture patterns inferred by *PCAngsd* or if the data is better explained by alternative scenarios such as recent bottlenecks, we used chromosome painting and patterns of allele sharing to construct painting palettes via the programmes *MixPainter* and *badMIXTURE* (unlinked model)[28] and compared this to the *PCAngsd*-inferred palettes (Fig. 2B, C; Supplementary Methods S1.2). We referred to patterns of residuals between these palettes to inform of the most likely underlying demographic scenario. For assessing Alpine–Balkan palette residuals (and hence admixture), 65 individuals each from the French Alps (inferred as pure Alpine ancestry in PCAngsd), Monte Baldo (inferred with both Alpine and Balkan ancestries in *PCAngsd*) and Julian Alps (inferred as pure Balkan ancestry in *PCAngsd*) were analysed under $K = 2$ in *PCAngsd* and *badMIXTURE* (Fig. 2C). For assessing Apennine–Balkan admixture, 22 individuals each from the French pre-Alps (inferred as pure Apennine ancestry in *PCAngsd*), Tuscany (inferred with both Apennine and Balkan ancestries in *PCAngsd*) and Julian Alps (inferred as pure Balkan ancestry in *PCAngsd*) were analysed under $K = 2$ in *PCAngsd* and *badMIXTURE*.

To construct a genetic distance tree (Supplementary Fig. S1), we first calculated pairwise genetic distances between 549 individuals (5 individuals per population for all populations) using *ATLAS*, employing a distance measure (weight) reflective of the number of alleles differing between the genotypes (Supplementary Methods S1.2; Supplementary Data 1). A tree was constructed from the resultant distance matrix via an initial topology defined by the BioNJ algorithm with subsequent topological moves performed via Subtree Pruning and Regrafting (SPR) in *FastME* version 2.1.6.1[62]. This matrix of pairwise genetic distances was also used as input for analyses of effective migration and effective diversity surfaces in *EEMS*[25]. *EEMS* was run setting the number of modelled demes to 1000 (Fig. 2A, Supplementary Fig. S8). For each case, ten independent Markov chain Monte Carlo (MCMC) chains comprising 5 million iterations each were run, with a 1 million iteration burn-in, retaining every 10,000th iteration. Biogeographic barriers (Fig. 2A, Supplementary Fig. S7) were further identified via applying Monmonier's algorithm[24] on a valuated graph

constructed via Delauney triangulation of population geographic coordinates, with edge values reflecting population pairwise $F_{ST}$; via the *adegenet* R package version 2.1.1[63]. $F_{ST}$ between all population pairs were calculated via *ANGSD*, employing a common sample size of 5 individuals per population (Supplementary Fig. S6; Supplementary Methods S1.2; Supplementary Data 1). 100 bootstrap runs were performed to generate a heatmap of genetic boundaries in space, from which a weighted mean line was drawn (Supplementary Fig. S7). All analyses in *ANGSD* were performed with the GATK (-GL 2) model, as we noticed irregularities in the site frequency spectra (SFS) with the SAMtools (-GL 1) model similar to that reported in Ref. [58] with particular BAM files. All analyses described above were performed on the full genome.

## Ancestral sequence reconstruction

To acquire ancestral states and polarise site-frequency spectra for use in the directionality index $\psi$ and demographic inference, we reconstructed ancestral genome sequences at each node of the phylogenetic tree of 9 *Dianthus* species: *D. carthusianorum*, *D. deltoides*, *D. glacialis*, *D. sylvestris* (Apennine lineage), *D. lusitanus*, *D. pungens*, *D. superbus alpestris*, *D. superbus superbus*, and *D. sylvestris* (Alpine lineage). This tree topology was extracted from a detailed reconstruction of *Dianthus* phylogeny based on 30 taxa by Fior et al. (Fior, Luqman, Scharmann, Zemp, Zoller, Pålsson, Gargano, Wegmann & Widmer; paper in preparation) (Supplementary Methods S1.3). For ancestral sequence reconstruction, one individual per species was sequenced at medium coverage (ca. 10x), trimmed (*Trimmomatic*), mapped against the *D. sylvestris* reference assembly (*BWA-MEM*) and had overlapping read pairs clipped (*bamUtil*) (Supplementary Methods S1.3). For each species, we then generated a species-specific FASTA using *GATK* FastaAlternateReferenceMaker. This was achieved by replacing the reference bases at polymorphic sites with species-specific variants as identified by *freebayes*[64] (version 1.3.1; default parameters), while masking (i.e., setting as "N") sites (i) with zero depth and (ii) that didn't pass the applied variant filtering criteria (i.e., that are not confidently called as polymorphic; Supplementary Methods S1.3). Species FASTA files were then combined into a multi-sample FASTA. Using this, we probabilistically reconstructed ancestral sequences at each node of the tree via *PHAST* (version 1.4) prequel[65], using a tree model produced by *PHAST* phylofit under a REV substitution model and the specified tree topology (Supplementary Methods S1.3). Ancestral sequence FASTA files were then generated from the prequel results using a custom script.

## Expansion signal

To calculate the population pairwise directionality index $\psi$ for the Alpine lineage, we utilised equation 1b from Peter and Slatkin (2013)[31], which defines $\psi$ in terms of the two-population site frequency spectrum (2D-SFS) (Supplementary Methods S1.4). 2D-SFS between all population pairs (10 individuals per population; Supplementary Data 1) were estimated via *ANGSD* and *realSFS*[66] (Supplementary Methods S1.4), for unfolded spectra. Unfolding of spectra was achieved via polarisation with respect to the ancestral state of sites defined at the *D. sylvestris* (Apennine lineage) - *D. sylvestris* (Alpine lineage) ancestral node. Correlation of pairwise $\psi$ and (great-circle) distance matrices was tested via a Mantel test (10,000 permutations). To infer the geographic origin of the expansion (Fig. 3), we employed a time difference of arrival (TDOA) algorithm following Peter and Slatkin (2013)[31]; performed via the *rangeExpansion* R package version 0.0.0.9000[31,67]. We further estimated the strength of the founder of this expansion using the same package.

## Demographic inference

To evaluate the demographic history of *D. sylvestris,* a set of candidate demographic models was formulated. To constrain the topology of

tested models, we first inferred the phylogenetic tree of the three identified evolutionary lineages of *D. sylvestris* (Alpine, Apennine and Balkan) as embedded within the larger phylogeny of the Eurasian *Dianthus* clade (note that the phylogeny from Fior et al. (Fior, Luqman, Scharmann, Zemp, Zoller, Pålsson, Gargano, Wegmann & Widmer; paper in preparation) excludes Balkan representatives of *D. sylvestris*). Trees were inferred based on low-coverage whole-genome sequence data of 1–2 representatives from each *D. sylvestris* lineage, together with whole-genome sequence data of 7 other *Dianthus* species, namely *D. carthusianorum*, *D. deltoides*, *D. glacialis*, *D. lusitanus*, *D. pungens*, *D. superbus alpestris* and *D. superbus superbus*, that were used to root the *D. sylvestris* clade (Supplementary Methods S1.5). We estimated distance-based phylogenies using *ngsDist*[68] that accommodates genotype likelihoods in the estimation of genetic distances (Supplementary Methods S1.5). Genetic distances were calculated via two approaches: (i) genome-wide and (ii) along 10 kb windows. For the former, 110 bootstrap replicates were calculated by re-sampling over similar-sized genomic blocks. For the alternative strategy based on 10 kb windows, window trees were combined using *ASTRAL-III* version 5.6.3[69] to generate a genome-wide consensus tree accounting for potential gene tree discordance (Supplementary Methods S1.5). Trees were constructed from matrices of genetic distances from initial topologies defined by the BioNJ algorithm with subsequent topological moves performed via Subtree Pruning and Regrafting (SPR) in *FastME* version 2.1.6.1[62]. We rooted all resultant phylogenetic trees with *D. deltoides* as the outgroup[70]. Both approaches recovered a topology with the Balkan lineage diverging prior to the Apennine and Alpine lineages (Supplementary Fig. S9). This taxon topology for *D. sylvestris* was supported by high *ASTRAL-III* posterior probabilities (>99%), *ASTRAL-III* quartet scores (>0.5) and bootstrap values (>99%). Topologies deeper in the tree were less well-resolved (with quartet scores <0.4 in more basal nodes). Under the inferred *D. sylvestris* topology and a less-assumptive simultaneous trichotomous split topology, 18 models were formulated spanning from simple to complex (Supplementary Fig. S10). Complex models allowed for population size changes and different migration rates (which could further be asymmetric) at each time epoch. We allowed up to five time epochs to accommodate (i) the two divergence events, (ii) the bottleneck-like effect of contemporary sampling, and (iii) up to two additional transitions in demography.

To estimate the demographic parameters of these models, we used *moments* version 1.0.0[71] to evaluate populations' joint site frequency spectra. We estimated the unfolded three-population joint site frequency spectrum (3D-SFS) comprised of one representative population each per lineage via *ANGSD* and *realSFS*, using 20 individuals per population (Supplementary Methods S1.5; Supplementary Data 1). The spectra were polarised with respect to the ancestral state of sites defined at the *D. lusitanus* - *D. sylvestris* (Alpine lineage) ancestral node (for tree topology, see Supplementary Methods S1.3). To facilitate model selection and optimisation in *moments*, we employed an iterative optimisation procedure, modified from Ref. [72] (Supplementary Methods S1.5). Model selection was performed via comparison of model log-likelihood values, the Akaike information criterion (AIC) and via an adjusted likelihood ratio test based on the Godambe Information Matrix (GIM) (Supplementary Table S1). To estimate confidence intervals for demographic parameters, we employed a nonparametric bootstrapping strategy by generating 100 bootstraps of the 3D-SFS, resampling over unlinked genomic blocks. Parameter uncertainties were then calculated by fitting bootstrap datasets in moments under the described optimisation procedure (Supplementary Methods S1.5). To convert generation time to calendar years, we assume a generation time of three years as inferred from population growth models of a closely related species (*D. carthusianorum*) with similar life history (Pålsson, Walther, Fior & Widmer; paper in preparation).

## Distribution modelling

To model species and lineage distributions in space and time, we acquired species occurrence data and environmental data from various sources. Species occurrence data was acquired from Conservatoire Botanique National Méditerranéen de Porquerolles (CBNMed; http://flore.silene.eu), Conservatoire Botanique National Alpin (CBNA; http://flore.silene.eu), GBIF (https://www.gbif.org; https://doi.org/10.15468/dd.zzqdys), iNaturalist (https://www.inaturalist.org), Info Flora (https://www.infoflora.ch), Wikiplantbase #Italia (http://bot.biologia.unipi.it/wpb/italia), Sweden's Virtual Herbarium (http://herbarium.emg.umu.se), Virtual Herbaria Austria (https://www.jacq.org) and personal collaborators. The environmental data comprised an initial set of 19 bioclimatic variables (CHELSA[32]) together with 3 topographic variables (elevation, slope and aspect) (GMTED2010 and CHELSA PaleoDEM[32,73]), soil type and pH (at 5 cm depth) (SoilGrids[74]). Prior to running SDMs, a coherent set (consistent across all lineages) of most important, least collinear and biologically relevant variables was selected. Variable selection followed an iterative process of model fitting via generalised linear (GLM; *ecospat* R package version 3.0[75]), maximum entropy (maxent;[76] *dismo* R package version 1.1.4[77]) and random forest (RF; *randomForest* R package version 4.6.14[78], *extendedForest* R package version 1.6.1[36]) modelling to assess variable importance, combined with variance inflation factor (VIF; *usdm* R package version 1.1.18[79]) and correlation analyses. We retained the most important and least collinear variables (VIF < 10 and Pearson's correlation $r$ < 0.7), based on recommended cut-offs for this type of analysis[80]. This resulted in a final set of 10 variables: (1) isothermality, (2) temperature seasonality, (3) temperature maximum warmest month, (4) temperature mean wettest quarter, (5) temperature mean driest quarter, (6) precipitation seasonality, (7) precipitation warmest quarter, (8) precipitation coldest quarter, (9) soil pH at 5 cm and (10) topographic slope. Distribution models were generated for each lineage separately as well as for the pooled species. For each run, occurrences were randomly sampled from a larger set and disaggregated to balance sampling density in geographic space, resulting in ca. 420, 260, 170 and 530 retained occurrences for the Alpine, Apennine, Balkan and pooled species, respectively. Using these occurrence data and the final set of environmental predictors, we generated an SDM ensemble model built from the weighted average of four separate models: (1) a generalised linear model (GLM), (2) a general additive model (GAM), (3) a maximum entropy model (maxent) and (4) a random forest model (RF). Model weights reflected their classification performance (i.e., area under the curve (AUC) of the receiver operating characteristic (ROC) curve). For each model, 5-fold cross-validation and model evaluation was performed. The resultant ensemble SDM model was projected onto present-day climate as well as hindcasted to the LGM climate (Supplementary Methods S1.6). For the LGM environmental predictors, we took the ensemble (mean) of four PMIP3 global climate models (GCMs) implemented under CHELSA[32]: (1) NCAR-CCSM4[81], (2) MIROC-ESM[82], (3) MRI-CGCM3[83], and (4) MPI-ESM-P;[84] selecting models that are distinct and with low amount of interdependence between each other[85]. All environmental variables apart from soil variables were available for present-day and LGM predictor datasets. We thus assume in our models that soil pH remained constant through time. While this a strong assumption, the alternative strategy of excluding pH would similarly assume a constant effect in time, in addition to a constant effect across space. Here, we argue that it is better to have an informed constant over an uninformed constant; given that no paleo-model of global soil is currently available.

To reconstruct lineage-specific routes of expansion and assign present-day populations to their most likely ancestral refugia, we first projected lineage-specific niche models to the LGM climate. We identified distinct, contiguous refugia (spatial clusters of predicted occurrences) via an unsupervised density-based spatial clustering (DBSCAN) algorithm (*dbscan* R package version 1.1.2[86]), which

coincided with, and were denominated according to, geographic regions (Alps, Apennines and Balkans). We then sequentially projected the lineage-specific niche models to climate rasters at 100-year time intervals for the period between the LGM to the present-day. At each (successive) time-point, new occurrences (in space) were assigned a lineage based on the $k$-nearest neighbour ($k = 1$) of the previous (time-point's) set of lineage-assigned occurrences (*FNN* R package version 1.1.2.1[87]), conditional that the new occurrences lie within a defined distance $d$ away (dispersal parameter, $d = 12$ km per century[88]). In addition to this limit on dispersal rate, we enforced competitive exclusion between lineages such that only a single lineage can occupy a spatial cell at a given time. The 100-year time interval dataset was generated by linearly interpolating climatic variables between the current and ensemble LGM models. While it is known that climate did not change in a linear fashion between these time points, we considered this approach to be more informative than the alternative of assigning present-day occurrences to geographic refugia based on distance measures, as the former incorporates landscape heterogeneity in an explicit albeit approximative spatio-temporal manner. Recently, CHELSA-TraCE21k[89] was published which explicitly models climate in 100-year time intervals from the LGM to the present-day. However, LGM hindcasts from this model were inconsistent with that of the other 4 PMIP3 models employed here, potentially as a result of being based on the older CCSM3 GCM[90], and so we refrained from use of the TraCE dataset here.

## Visualising shifts in environment space and habitat availability

To visualise the shift in environmental space and habitat availability from LGM to present, we projected the environmental space of the LGM and present-day to a common, lower-dimensional space, via applying the PCA transformation (scaling, centreing and rotation) of the present-day environment to both present-day and LGM environments (Supplementary Methods S1.7). Here, the assessed environmental space comprised the 19 bioclimatic variables (CHELSA) together with elevation (GMTED2010, Chelsa PaleoDEM) and topographic slope. The density of cells occupying each coordinate in the resultant PCA-transformed environmental space was visualised via hexagonal binning (*ggplot2* R package version 3.3.2), which allowed for the quantification of area at each point (in the PCA-transformed environmental space). The geographic extents in which the environmental data were taken from and constrained to are shown in Supplementary Fig. S16.

## Predicting adaptive genetic structure in space and time

To predict the sieving of adaptive genetic variation in space and time, we modelled the association of genetic variants (SNPs) with changes in environment, using gradient forest (GF)[17,36]. Here, we assume that contemporary gene-environment associations distributed across space reflect gene-environment associations across time[9,17,20,21]. GF characterises compositional turnover in allele frequencies along environmental gradients via monotonic, non-linear (turnover) functions that transform multidimensional environmental space into multidimensional genomic space[17,36]. Starting with the full set of environmental variables described above, environmental predictors were selected by quantifying variable importance via random forests (*gradientForest* R package 0.1.18[36]) and by assessing variable collinearity via variance inflation factor (VIF) and correlation analyses. We retained the most important and least collinear variables (VIF < 10 and Pearson's correlation $r < 0.7$), resulting in a final set of 10 variables: (1) temperature diurnal range, (2) temperature seasonality, (3) temperature minimum coldest month, (4) temperature mean wettest quarter, (5) temperature mean driest quarter, (6) precipitation seasonality, (7) precipitation warmest quarter, (8) precipitation coldest quarter, (9) soil pH at 5 cm, and (10) topographic slope. To account for genetic structure in the Alps (Alpine lineage), we included longitude and

latitude as co-variates in the model, in light that these were shown to correlate strongly with the main two principal components of genome-wide structure (Supplementary Fig. S13). As an alternative method, we built a Moran's Eigenvector Map (MEM)[91] via the adespatial R package version 0.3.8[92] based on a spatial weighting matrix reflective of the Alpine lineage's expansion history, and included this as a co-variate in the GF model (instead of longitude and latitude) (Supplementary Fig. S21). Here, the edges of the spatial weighting matrix were weighted by the divergence of expansion paths (from the LGM refugia) between population pairs (see the path overlap and divergence metrics of van Etten & Hijmans (2010);[88] Supplementary Fig. S21, Supplementary Table S2). This measure of the divergence of expansion routes was calculated over a spatial transition matrix (i.e., resistance surface) defined by the lineage-specific SDM projection, via the gdistance R package version 1.2.2[93], utilising the random walk algorithm. Samples were considered as neighbours in the spatial weighting matrix if their pairwise geographic distance was equal to or less than the longest edge of the minimum spanning tree. Of the three positive MEM eigenvectors returned, a single positive eigenvector (MEM1) explained the majority of the variance and was used as the spatial co-variate (Supplementary Fig. S21). For the response variable, our GF model takes population allele frequencies. Genetic variants (SNPs) segregating across 43 Alpine populations comprising 14 individuals each were first identified using the programme *freebayes*[64] (version 1.3.1) (Supplementary Methods S1.8; Supplementary Data 1). We constrained the variant-callset to the exon regions of the genome. Population allele frequencies were calculated from the resultant VCF via *vcflib* popStats (version 1.0.1.1)[94] utilising genotype likelihoods. We filtered this dataset to retain only sites with depth ≥7 per population. GF was run on the resultant set of 390,262 exon SNPs in batches of 10,000 SNPs, using 500 decision trees. Batch runs were combined via combined-GradientForest {standardise = "before", method = 2} in the *gradientForest* R package. When taking the ensemble of all exon genetic variation—weighting the contribution of each SNP by the coefficient of determination ($R^2$) of its environment association (Supplementary Fig. S17)—our GF approach can potentially include the contribution (effects) of a major fraction of adaptive loci, including those of small effect size that are affected by polygenic selection. Recent evidence has demonstrated that such an approach based on large sets of genomic SNPs can reflect fitness well[19], performing on par or better that GF models based on a priori identified environmentally-associated SNPs[19]. The resultant GF turnover functions—which transform the environmental variables (environmental space) to biological variables of composition turnover (biological space)[17,36]—are applied to present-day and LGM climate rasters, to characterise adaptive genomic composition of populations in space and time. To visualise adaptive genomic composition in space, we plotted the first three principal components of the transformed environmental space excluding the transformed longitude and latitude variables, to visualise only the adaptive component. Note here that the PCA was centred but not scaled to retain GF-calculated importance of the transformed environmental variables[17,19].

To evaluate differences in genomic composition and quantify evolutionary change of populations between assessed time points, we evaluated (genomic) compositional differences (between time points) accounting for the location of the populations' glacial refugia. Specifically, we calculated the multivariate Euclidean distance between the genomic composition of every population in the present-day and that of its (geographically) closest predicted refugial source at the time of the LGM (Supplementary Fig. S20). We term this metric "glacial genomic offset". Here, in contrast to our visualisation of adaptive genomic composition earlier, we retain the biological variables transformed from longitude and latitude in our calculation of the glacial genomic offset to incorporate the effect of IBD and distance-associated drift processes (i.e., expansion). This is because we aim

**Article**

for the glacial genomic offset to encapsulate the joint, realised effects of isolation by environment, isolation by distance (IBD) and spatial expansion between the ancestral population in the refugia and the present-day population. For our alternative method using MEM, we interpolated MEM values via inverse path distance weighting across the SDM-defined resistance surface via the ipdw R package version 0.2.6[95], to include (model) the effect of distance in the glacial genomic offset. Such an interpolation approach is coarse and prone to artefacts; however, provides an alternative way of modelling distance effects when genetic structure is not well-represented by a geographic cline. Note that in our calculation of glacial genomic offsets, the geographic extent of glacial refugia relied on distribution models, which assumes niche conservatism. This may appear counter to the aim here of capturing adaptation. We alleviate this methodological constraint, however, by employing lineage-specific, rather than species, niches and by relying additionally on population genetic inferences ($\psi$) to reconstruct glacial refugia. Moreover, our inference of past distributions are conservative, such that inferred glacial refugia would have been smaller, not larger, had adaptation facilitated post-glacial expansion. Finally, we note that our prediction of adaptive variation during the LGM is highly homogenous across space, meaning that our predictions of glacial genomic offset remain robust under fluctuations in the inferred geographic extent of glacial refugia.

To explore the adaptive relationship between low-elevation individuals, high-elevation individuals and refugial-proxies (i.e., those presently inhabiting the inferred glacial refugia in Monte Baldo and the western Dolomites), we partitioned populations (excluding refugial-proxies) into low-elevation (<1000 m) and high-elevation (>1500 m) categories. To avoid biases related to imbalanced sample sizes, categories were sub-sampled to a common sample size of 70 individuals (5 populations) each (Supplementary Data 1). We then calculated a genetic distance tree, PCA and Venn diagram of allele presence and absence; based on the top (unlinked) 1000 GF environmentally-associated SNPs (Supplementary Fig. S19). The genetic distance tree and PCA were calculated as described above for the whole-genome dataset. For allele presence-absence, we applied a minimum allele frequency threshold of 5%.

## Validation of glacial genomic offset

To validate our predictions of glacial genomic offset, we performed correlations of this metric with various population genetic diversity and neutrality statistics including nucleotide diversity $\pi$, Tajima's $D$, Fu & Li's $F$, Fay & Wu's $H$, and Zeng's $E$. Statistics were calculated for sampled populations both genome-wide and centred around environmentally-associated loci, using *ANGSD* (Supplementary Method S1.9). For the latter, we calculated the weighted mean statistics of exon SNPs with weights given by the coefficient of determination ($R^2$) of the SNP's environmental association (as given under GF; Supplementary Fig. S17A). This approach avoids the lossy strategy of calling discrete adaptive candidates and potentially better reflects genome-wide (including polygenic) signals of adaptive diversity. We further compared levels of nucleotide diversity ($\pi$) for contemporary low- and high-elevation populations (ca. 1000 m elevation difference) in areas where they co-occur in close geographic proximity, to control for the effects of isolation by distance (IBD) and the spatial expansion (four pairs total; Supplementary Data 1). Given the low number of suitable population pairs, we additionally compared $\pi$ for all populations partitioned into low-elevation (<1000 m) and high-elevation (>1500 m) bins, ordered along the expansion axis. $\pi$ was calculated both genome-wide ($\pi_{GW}$) and centred around environmentally-associated loci ($\pi_{GF}$). The latter was calculated as the weighted mean $\pi$ of exon SNPs, with weights given by the $R^2$ of the SNP's environmental association (as given under GF; Supplementary Fig. S17B).

## Access of samples

Access, collection and import/export of samples were conducted in a responsible manner and in compliance with all relevant local, national and international laws. All necessary sampling permits were obtained prior to sample collection. This comprised of sampling permits from Comunità della Vallagarina for sampling in Monte Baldo, Italy (issued on 13 July 2017, valid for the year 2017); from Parco Nazionale Gran Paradiso for sampling in Gran Paradiso National Park, Italy (issued on 12 May 2017, valid for the summer of 2017); from Amt für Natur und Umwelt for sampling in Graubünden, Switzerland (issued on 21 April 2017, valid for the years 2017 and 2018); from Amt für Natur, Jagd und Fischerei for sampling in St. Gallen, Switzerland (issued on 29 May 2017, valid from the date of issue until 31 December 2018), and from Amt für Wald und Landschaft for sampling in Obwalden, Switzerland (issued on 29 May 2017, valid for May-September 2017 and May-September 2018). Nagoya Protocol on Access and Benefit Sharing was followed in countries where it had been ratified at the time of sampling. *Dianthus sylvestris* is not listed under the Convention on International Trade in Endangered Species of Wild Fauna and Flora (CITES) or under the International Union for Conservation of Nature (IUCN) Red List of Threatened Species.

## Reporting summary

Further information on research design is available in the Nature Portfolio Reporting Summary linked to this article.

## Data availability

Raw sequencing reads for the 1261 low-coverage *D. sylvestris* whole genomes are deposited and available at the European Nucleotide Archive (ENA) under accession code PRJEB53522. Raw sequencing reads for the across-species dataset are available at ENA under accession code PRJEB54098. The *D. sylvestris* genome reference assembly and structural annotation are available at the Dryad repository: https://doi.org/10.5061/dryad.x0k6djhng[54]. Sample accessions and metadata are provided in Supplementary Data 1. Environmental data used was downloaded from CHELSA (version 1.2; https://chelsa-climate.org) and SoilGrids (2020 version; https://soilgrids.org). Topographic data was downloaded from CHELSA and GMTED2010 (2010 version; https://topotools.cr.usgs.gov/gmted_viewer/gmted2010_global_grids.php).

Species occurrence data was acquired from Conservatoire Botanique National Méditerranéen de Porquerolles (CBNMed; http://flore.silene.eu), Conservatoire Botanique National Alpin (CBNA; http://flore.silene.eu), GBIF (https://www.gbif.org; https://doi.org/10.15468/dd.zzqdys), iNaturalist (https://www.inaturalist.org), Info Flora (https://www.infoflora.ch), Wikiplantbase #Italia (http://bot.biologia.unipi.it/wpb/italia), Sweden's Virtual Herbarium (http://herbarium.emg.umu.se), Virtual Herbaria Austria (https://www.jacq.org) and personal collaborators in 2017. This occurrence data is deposited at the Github repository (https://github.com/hirzi/RhEA; https://doi.org/10.5281/zenodo.7581797)[96].

## Code availability

Code for performing demographic inference, running distribution models, visualising shifts in environmental space, performing gradient forest and calculating glacial genomic offsets are available at the GitHub repository: https://github.com/hirzi/RhEA (https://doi.org/10.5281/zenodo.7581797)[96].

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

## Acknowledgements

We would like to thank Claudia Michel for her invaluable work in the lab. We are grateful to Felix Gugerli, who provided helpful comments in the early stages of this study, and to Loïc Pellissier, whose feedback aided our implementation of the SDMs. We thank Karsten Rohweder, Bostjan Surina and Salvatore Cozzolino for providing additional plant samples, and Ivana Rešetnik and Martina Temunovic, for sharing occurrence records for the Balkans. We acknowledge the Conservatoire Botanique National Méditerranéen de Porquerolles, Conservatoire Botanique National Alpin, GBIF, iNaturalist, Info Flora, Sweden's Virtual Herbarium, Virtual Herbaria Austria and Wikiplantbase #Italia for sharing further species occurrence data. We are grateful to Domenico Gargano for assisting in plant identification, and to the Genetic Diversity Centre at ETH Zurich and in particular Niklaus Zemp for providing IT support. This work was supported by the Swiss National Science Foundation (SNSF) grants 31003A_160123 and 31003A_182675 awarded to A.W., and grant 31003A_173062 awarded to D.W.

## Author contributions

H.L., A.W., and S.F. designed the study. H.L. collected samples from the field and performed all analyses. D.W. helped with methods. H.L. wrote the manuscript, which all authors revised.

## Competing interests

The authors declare no competing interests.
