## [Peer Review File · Nature Communications]

Climate-induced range shifts drive adaptive response via spatio-temporal sieving of allelesReviewers' Comments:

Reviewer #1:

Remarks to the Author:

The manuscript by Luqman et al. aims at testing whether it is possible to find a genomic signature coherent with adaptive responses during post-glacial range shift. While migration (range shift) has been extensively recognized as the main response to climate change for more than three decades (Huntley & Webb 1989), recent papers have challenged this paradigm through conceptual arguments suggesting that adaptation likely occurred along with range shift (Davis & Shaw 2001, de Lafontaine et al. 2018). So far, this paradigm shift has not been tested empirically, which is where the paper by Luqman et al. comes into play. The present manuscript relies on an impressive amount of data analyses, yielding robust results all pointing in the same direction. Together, they provide a comprehensive empirical assessment of the theory that the interplay of adaptation and range shifts has been central in species responses to Quaternary climatic change. This provides key insights about how species are likely to respond to ongoing anthropogenic climate change. As such it represents a timely, high impact, contribution to our understanding of eco-evolutionary dynamics.

I only have a few minor comments to help improve clarity.

Throughout the manuscript, the authors refer to "spatial sorting" or "spatio-temporal sorting". I strongly advise using another phrase. The term "spatial sorting" was coined by Shine et al. (2011) to refer to a spatial analogue of natural selection occurring specifically during range expansion. "Spatial sorting" (Shine et al. 2011) generates novel phenotypes (through sorting of alleles) that are adept at rapid dispersal, irrespective of how the underlying genes affect an organism's survival or its reproductive success. While the present paper also deals with evolution during range expansion, it goes beyond the definition of "spatial sorting" (Shine et al. 2011) as it is not limited to dispersal traits (alleles) –although it might include such alleles– and it does actually imply natural selection (of traits/alleles affecting an organism's survival or its reproductive success) during migration. As such, I strongly suggest to avoid using the term "spatial sorting". The term "spatio-temporal sorting" is only slightly better because it might still be misleading. At the end of the day, we really want to avoid introducing the same terminology to invoke distinct evolutionary processes occurring during range expansion, one of which is limited to dispersal traits and involves no natural selection (i.e. no adaptation), which is in strike contrast with the present effort actually suggesting such adaptation did occur during range expansion.

Figure 1. This conceptual figure is important but need to be slightly revised. 1) Please use monochrome color to fill the curves of individual genotype's niche. The use of multiple colors and design within each curve becomes very confusing, especially where they overlap (see chartjunk -Tuftes 2001). I advise opting for monochrome colors with some level of transparency so one can easily see the overlaps. 2) I am uncertain whether my interpretation of the figure is entirely correct, but in figure 1C I would see a decline in allele/genotype frequency if the curved refer to the second 'blue' genotype (starting from the left) instead of the third 'wavy grey' genotype. This implies that T1 should be moved towards the left of the graph (around the mode of the second 'blue' genotype). It becomes misleading because in the lower 'interglacial' graph, the mode of the realized niche matches the mode of the third 'wavy grey' genotype, giving the impression that the niche of this specific genotype exactly matches the core of its environmental niche during the interglacial. In such case why would his frequency drop during the interglacial. Then again, maybe my interpretation of the figure is wrong but then, I advise that you clarify because such a conceptual figure usually aims at an being understood quite readily.

Specific comments

Line 39. The paper by Hewitt (2000) is really more about the genetic consequences of the Milankovitch cycles. However, the sentence does introduce this topic. I suggest replacing (or at least

add) Webb & Bartlein (1992).

Line 75-77 I think perhaps this assumption needs to be unpacked, as it is key to the genomic offset prediction.

Line 117, lines 394-396, Figure 2 and Figure S8. Please clarify the use of EEMS, as it varies throughout the text: reflect population effective diversity rates (q), elevated genetic differentiation. Also the result of EEMS is shown twice (Figure 2A and S8) but the two patterns are different. It is unclear how the analysis differs between these two, given that the method only refer to a single EEMS run. It also seems like there might be some overfitting in Figure S8.

Lines 326-329. I am not sure I fully agree with lines 326-329. Results suggest that the populations inhabiting warm elevation environment have shown their capacity to adapt to these warmer condition. While a selective sweep might have reduced adaptive variation, this was likely towards 'sieving' for increasing the frequency of those alleles responsible for a better fitness in warmer environments. Hence directional selection (under warmer climate) should keep on 'sieving' in this same direction. Should we assume that populations have already 'exhausted' their adaptive potential to warmer environments? Furthermore, the conceptual model shown in figure 1C suggests antagonistic pleiotropy as the main mechanism (a genotype/allele is either adapted or maladapted depending on the environment) but it is also possible that some alleles were adaptive in the past but became invisible to selection (neutral) in modern day conditions (temporal conditional neutrality - de Lafontaine et al. 2018). Such alleles would still bear adaptive potential but simply shifted to a neutral state in contemporary environments. Because they follow the neutral background, they are not expected to drastically decrease in frequency as much as maladapted alleles (when assuming antagonistic pleiotropy) during a sweep. The corollary is that some currently neutral alleles could eventually become visible to selection as climate changes. Hence there would still have plenty of room for future adaptation. I thus disagree with the assumption that low elevation population became more vulnerable, but they certainly are populations worth investigating to assess how marginal populations might bear unique alleles/genotypes/traits conferring fitness advantages that might provide rapid resilience to the species in a warmer climate!

I truly hope these few comments above will at least provide some food for thought!

Best regards,

Guillaume de Lafontaine

Literature cited

Davis, M. B., & Shaw, R. G. (2001). Range shifts and adaptive responses to Quaternary climate change. *Science*, 292, 673–679.

de Lafontaine, G., Napier, J. D., Petit, R. J., & Hu, F. S. (2018). Invoking adaptation to decipher the genetic legacy of past climate change. *Ecology*, 99(7), 1530–1546.

Hewitt, G. M. (2000). The genetic legacy of the Quaternary ice ages. *Nature*, 405, 907–913.

Huntley, B., & Webb III, T. (1989). Migration: Species' response to climatic variations caused by changes in the earth's orbit. *Journal of Biogeography*, 16, 5–19.

Shine, R., Brown, G. P., & Phillips, B. L. (2011). An evolutionary process that assembles phenotypes through space rather than through time. *Proceedings of the National Academy of Sciences of the United States of America*, 108(14), 5708–5711.

Tufte, E. R. (2001). *The Visual Display of Quantitative Information* 2nd edition, Graphics Press, Cheshire, CT

Webb III, T., & Bartlein, P. J. (1992). Global changes during the last 3 million years: Climatic controls and biotic responses. *Annual Review of Ecology and Systematics*, 23, 141–173.

Reviewer #2:

Remarks to the Author:

How climate change influences population demography and local adaptation across a species' range has become a huge focus of research. The current study looks at historic changes in climate niche and genomic composition since the LGM to ask whether sorting of climate-adaptive genetic variation occurred during population range shifts away from glacial refugia.

The study represents a sophisticated integration of landscape genomics and climate modeling whose novelty primarily lies with the elegance by which the various analyses come together to address how adaptation proceeded from standing variation during the expansion out of refugia. At some level this result is expected as well as consistent with theory and a wealth of empirical work demonstrating contemporary clines in phenotypes or alleles that are unlikely to have arisen purely from de-novo mutation. However, few studies have produced as large a population-level dataset of whole-genome sequences and related neutral and adaptive components of genomic diversity to the probable location of glacial refugia. Further, none that I'm aware of have shown that genomic composition at climate-associated alleles bear signatures of selection that are proportional to the inferred magnitude of shifting gene-environment relationships (genomic offset). The latter derives from the authors' derivation of a new genomic offset metric that focuses on differences between contemporary and glacial refugial climate, as opposed to most offset studies which focus on current and future climates.

Overall, I found the study very interesting, well written, and meticulously analyzed. I have a few major comments, followed by mostly minor suggestions:

1) I think that the idea behind glacial genomic offset is quite interesting, but I'm not convinced that the approach as implemented captures the signals from both adaptation and IBD/expansion. The formulation calculates pairwise offsets based on refugial climate and soil variables (makes sense), but I don't think that latitude and longitude are good proxies for IBD/expansion (L. 580-583), as they ignore barriers to movement/occupancy and are also inherently correlated with other aspects of the environment. Keeping with the same framework of relating contemporary populations to the predicted refugial site when calculating offsets, you could make use of Moran's Eigenvector Matrices (MEMs) to represent spatial connectivity between your sample populations and the refugium that may better represent expansion history. The edges in the network can be weighted to incorporate knowledge about connectivity (e.g., geographic distance or perhaps other isolating features of topography, habitat suitability, or even better by incorporating routes of expansion from your lineage-specific SDMs). Without a more sophisticated incorporation of the effects of landscape distance and connectivity, I think statements that glacial genomic offset "explicitly integrate the effects of past adaptation, migration, and expansion" (L. 296) are too far-reaching.

2) In projecting offsets back to the LGM, it seems that you're also extrapolating beyond the range of climate values represented by contemporary populations used to train the GF model. This climate novelty is apparent in Fig 4B. It's not known how this extrapolation is likely to bias offset estimates, and this point should be acknowledged and discussed for how it may impact your inference.

Introduction: In my opinion, the second paragraph sets up a bit of a false dichotomy, specifically where the argument is developed that post-Quaternary migration with no adaptation was/is the assumption, and that only recently has this paradigm been challenged. While Davis and Shaw (2001)

was a landmark paper that made the assertion that adaptation should be considered as part of the post-glacial process, that paper is now >20 years old, and a huge literature now focuses on adaptation during range expansions. I would suggest toning this down – still make the point, but don't force this as a recent paradigm shift. Here also I think the concept of selection on standing genetic variation (SGV) could be usefully introduced to complement the "sieving" analogy.

Minor comments:

L 88: "leverage on the expectation" – awkward phrasing

L. 128: I think most readers will interpret the phrase "strong reproductive isolation" to mean the presence of intrinsic (genetic) barriers to mating. But the lack of admixture you've shown here could be due to several different causes. Perhaps better to say, "a prolonged history of isolation".

L. 185-186: Isn't it circular to conclude that your lineage-specific SDMs generate distributions highly congruent with the genetic structure? The occurrences feeding into these SDMs are being defined by the genetic structure of these lineages.

L. 188: It seems to me that competitive exclusion between your lineages is more of an assumption you're making (out of convenience) in the modeling than a result that has statistical support. If you do want to show the latter, it would seem to require some sort of model selection process among variables that do/don't include that assumption.

L. 227: It's a misinterpretation to refer to "the adaptive genotype" when interpreting the visualization of genomic composition in Fig 5. These maps represent ordinations of population-level allele frequencies from across the genome. See also L. 234-235 ("predicted genotype").

L. 246: I quite like this set of analyses relating glacial genomic offset to π and the frequency of derived alleles, but I wouldn't consider this "validation". Rather, I think these analyses corroborate the offsets, but do not independently validate them.

L. 263-264 and Fig 6B: This is a nice result (higher nucleotide diversity at env-associated loci) but lacks statistical assessment. Can you include a Mann-Whitney test here between sets of loci? I would also temper the conclusion about highly diverged haplotypes (that hasn't specifically been shown). Lastly, it would be helpful to see the distributions of π values (e.g., in violin plots) rather than the point estimates in Fig 6B.

L. 356: Will the Fior et al. manuscript that describes the reference assembly used here for mapping be available prior to publication of the current paper? If not, consider making the genome assembly itself publicly available in a curated repository.

L. 364: Here, and throughout, "principle" should be "principal" when referring to PCA.

Fig 1B-C: I'm not sure about the patterned fill on these distributions...I think simple solid colors with partial transparency would be less distracting.

Supplemental:

L. 88-89: Did you hard-call genotypes prior to phasing?

Fig S16: What does the shading represent in this map? DEM?

Reviewer #3:

Remarks to the Author:

Lugman et al. use an impressive genomic database (whole genome resequencing in over 1200 individual & 115 populations) to analyze the evolutionary history of *Dianthus sylvestris* populations. With this data, the authors show that both adaptive and demographic processes have influenced the current distribution and shaped the genetic diversity of this species. Using clustering algorithms, species distribution models and demographic inference, the authors show that three lineages diverged during the PGIP, and remain well differentiated after having independent histories (three distinct refugia & low gene flow between groups). Second, the authors show that the Alpine populations had a recent expansion from their refuge, and that most populations grow in conditions similar to their respective refuge. They also show that the rest of the populations grow at lower altitudes and different conditions, suggesting they might have adapted to those conditions (by the sieving of alleles). Third, they find that populations growing at lower altitudes or at expansion routes have a higher "glacial genomic offset", suggesting that IBD and adaptation have shaped the genetic composition of populations. Importantly, the authors find that higher glacial genomic offsets are consistent with increased selective sweeps in low altitude populations, supporting their hypothesis. The authors conclude that the distribution and genetic diversity of *Dianthus sylvestris* is explained by a combination of migration and "sieving" of adaptive variation.

I think this manuscript is very interesting, and tests a very important question in evolutionary biology: How do populations respond to climate change? In short, their results suggest that populations can respond rapidly by migrating to suitable environments and simultaneously adapting to locations where they can tolerate new environments. I also think that the manuscript is very well written and that the methods they use are very complete and robust (although I have a few suggestions, see below). For example, I really like that the authors combined adaptive and neutral genetic diversity with environmental data in their analyses. I also think that the results are very robust, since the authors considered plenty of factors that could generate false positives or biases in their analyses (i.e., consider number of individuals sampled in clustering analyses; comparing admixture and chromosome painting tests; consider geographic patterns of isolation and colonization; using different algorithms to perform species distribution models; using different climatic circulation models; testing consistency between methods; comparing neutral vs climate related genes; corroborating glacial genomic offset with SFS deviation tests; contrasting altitude patterns vs geographic patterns; discussing the limitations of analyzing glacial genomic offsets using SDMs and assuming niche conservatism, etc.). Finally, I think they discuss critically many of the limitations in their analyses using important literature.

In sum, I think this is a very interesting paper, very well written and methodologically very robust. I think this paper is suited for Nature communications since it will be of interest for a very broad range of scientist.

Below, I address a few simple comments that I think would improve the paper.

1) Sieving hypothesis: I really like this hypothesis, and I think it is very relevant to combine adaptive and demographic data to understand evolution. In lines 61-63 & 69-71, the authors explain the hypothesis. However, I do not think that the hypothesis is explained clearly which affects the punch of the introduction. Also, I think that Figure 1 is not very intuitive. I am not a native English speaker (although I am fluent), and I did not know the word "Sieve". Until I searched for the definition, I was able to finally understand the hypothesis well. I think that a clearer explanation of the hypothesis would benefit the flow of the manuscript. Perhaps explaining step by step the sieving mechanism (migration first and selection second acting on adaptive alleles) might be helpful. I also think that Figure 1 could be simplified by adding a graphic representation of the mechanism of sieving of a population moving across a landscape (a simple schematic image). I also think that the shapes in the density plots add unnecessary complexity and are not esthetic. I would change to simple colors (I like the palette in Figure 2A) or shapes of grey.

2) Following this idea, I was puzzled that the authors mention the importance of allele sieving, but I did not find any result that explicitly explores the sieving of those alleles. The authors use the GF loci

to compare environmental related SNPs vs genome-wide SNPs. I get that the authors consider that analyzing the entire contribution of environmental related SNPs can show genome wide patterns of adaptation. I also get that there are low number of suitable pairs (1 606-607). However, I think it would be very interesting to perform more explicit analyses. For instance, the authors could do a genome environmental association analysis and detect selective sweeps that segregate in populations growing in alpine and low-altitude populations. In figure 3C the authors show that altitude does not correlate with geography suggesting that there are multiple altitude gradients and perhaps parallel evolution. The authors also find signals of selective sweeps in low altitude populations (Figure 6). I think it would be very interesting and valuable to test if there are alleles that in parallel are selected in those populations. It might be as simple as to test for environmental related GF alleles, if they correlate with altitude but not geography. Perhaps a NJ tree with adaptive alleles would group lowland populations and this could be contrasted with the genome-wide effects. If shared alleles occur in populations at different locations, It could indicate that adaptation from standing genetic variation is occurring.

3) Glacial genetic offsets: I think these methods and results are really interesting and smart. These results are important because they suggest that genetic offsets are powerful approximations to detect how populations will adapt to changing environments. In general, most studies that have used genetic offsets have the limitation that corroborating the results is complex. Only a few studies have included experimental validations. By analyzing patterns from past to present, the authors are able to test their results based on the validation of genetic diversity patterns (perturbations of SFS analyses). Having said that, I think that the results could be presented better. First, I think that for people who are not familiarized with genetic offsets, it would be convenient to present a schematic figure of how glacial genetic offset from refuge to current locations were estimated (the methods are very well written, but a schematic representation might help). Glacial genetic offset is a complex concept and this is a journal for a broad readership. Second, I agree that the results indicate that the current distribution of alpine populations has a similar projected genetic composition than the refuge in the past. However, there is no statistical analysis or metric that truly shows this. This is a problem, because the authors also support their sieving hypothesis by saying that low altitude locations grow at different climatic distributions. However, since the LGI refuge is so small it is not easy to analyze the different colors that are present in the refuge. For example, it is not clear if the "green-bluesish green" distribution did not exist in the past. I think that a good way to compare genetic compositions between present and past distributions (black polygons), would be to replicate the byplots in Figure 5A and only show the genetic composition distribution for locations within the polygons. This way it would be possible to see if in the byplot, the present alpine distribution overlaps with the refuge distribution; and to see if lowland populations also overlap with some of the past distribution. Finally, the authors could also differentiate between high and low altitude populations and use either centroids or distribution of PC scores to compare if the genetic compositions are significantly differentiated from the refuge genetic composition.

4) Why didn't the authors perform the glacial genomic offset analyses for the other genetic groups? This could go in supporting information. Is it because their potential distributions have been more conserved?

5) The authors use the names of geographic locations to point some results (i.e., Monte Baldo, Dolomites, etc.), however, these names are not present in the maps. For people that are not European, or not familiarized with these locations, it is a bit hard to follow. I think it would be convenient to add some of these names in the Figures or see if it is possible to only use North, South, East, West.

6) I think it would be convenient to make a supporting figure that shows the overlaps between the LGI and present Sdms of figure 4A. I would make one for each genetic group. This would help visualize the possible refugia. For the Alpine group, this is clear, but for the other two not so much.

7) I would like to see a short description or justification of why *Dianthus sylvestris* was used as a model.

8) How important is standing genetic variation for this hypothesis? I think this should be discussed more.

9) In figure 6 B & C, it is not very clear the distinction between nGF and nGW. At the right two square

brackets indicate this for Figure B and C but it is not easy to follow. Instead, I would change to colors indicating high and low altitude populations, and triangles and circles to indicate nGF and nGW, respectively.

RESPONSE TO REVIEWERS' COMMENTS

Reviewer #1 (Remarks to the Author):

The manuscript by Luqman et al. aims at testing whether it is possible to find a genomic signature coherent with adaptive responses during post-glacial range shift. While migration (range shift) has been extensively recognized as the main response to climate change for more than three decades (Huntley & Webb 1989), recent papers have challenged this paradigm through conceptual arguments suggesting that adaptation likely occurred along with range shift (Davis & Shaw 2001, de Lafontaine et al. 2018). So far, this paradigm shift has not been tested empirically, which is where the paper by Luqman et al. comes into play. The present manuscript relies on an impressive amount of data analyses, yielding robust results all pointing in the same direction. Together, they provide a comprehensive empirical assessment of the theory that the interplay of adaptation and range shifts has been central in species responses to Quaternary climatic change. This provides key insights about how species are likely to respond to ongoing anthropogenic climate change. As such it represents a timely, high impact, contribution to our understanding of eco-evolutionary dynamics.

I only have a few minor comments to help improve clarity.

Throughout the manuscript, the authors refer to “spatial sorting” or “spatio-temporal sorting”. I strongly advise using another phrase. The term “spatial sorting” was coined by Shine et al. (2011) to refer to a spatial analogue of natural selection occurring specifically during range expansion. “Spatial sorting” (Shine et al. 2011) generates novel phenotypes (through sorting of alleles) that are adept at rapid dispersal, irrespective of how the underlying genes affect an organism’s survival or its reproductive success. While the present paper also deals with evolution during range expansion, it goes beyond the definition of “spatial sorting” (Shine et al. 2011) as it is not limited to dispersal traits (alleles) –although it might include such alleles– and it does actually imply natural selection (of traits/alleles affecting an organism’s survival or its reproductive success) during migration. As such, I strongly suggest to avoid using the term “spatial sorting”. The term “spatio-temporal sorting” is only slightly better because it might still be misleading. At the end of the day, we really want to avoid introducing the same terminology to invoke distinct evolutionary processes occurring during range expansion, one of which is limited to dispersal traits and involves no natural selection (i.e. no adaptation), which is in strike contrast with the present effort actually suggesting such adaptation did occur during range expansion.

We appreciate this very helpful recommendation by the reviewer. To address the reviewer’s concern, we have rephrased “spatio-temporal sorting” to “spatio-temporal sieving” in the text (specifically in the title and in lines 29, 233, 326, 921 & 976).

Figure 1. This conceptual figure is important but need to be slightly revised. 1) Please use monochrome color to fill the curves of individual genotype’s niche. The use of multiple colors and design within each curve becomes very confusing, especially where they overlap (see chartjunk -Tuft 2001). I advise opting for monochrome colors with some level of transparency so one can easily see the overlaps. 2) I am uncertain whether my interpretation of the figure is entirely correct, but in figure 1C I would see a decline in allele/genotype frequency if the curved refer to the second ‘blue’ genotype (starting from the left) instead of the third ‘wavy grey’ genotype. This implies that T1 should be moved towards the left of the graph (around the mode of the second ‘blue’ genotype). It becomes misleading because in the lower ‘interglacial’ graph, the mode of the realized niche matches the mode of the third ‘wavy grey’ genotype, giving the impression that the niche of this specific genotype exactly matches the core of its environmental

niche during the interglacial. In such case why would his frequency drop during the interglacial. Then again, maybe my interpretation of the figure is wrong but then, I advise that you clarify because such a conceptual figure usually aims at an being understood quite readily.

We have now changed Figure 1 to comprise mainly of monochrome colours. The curves in Figure 1C indeed initially referred to the third and fourth curves (from the left) in Figure 1B (the genotype curves were to be interpreted in reference to the indicated temperature marks T1 and T2, rather than the glacial and interglacial curves). To avoid misinterpretation, we have shifted temperature marks T1 and T2 one curve to the left, to (mostly) overlap with the modes of the glacial and interglacial realized niches. Following comments from other reviewers, we have further added two new panels to illustrate the concept of sieving.

Specific comments

Line 39. The paper by Hewitt (2000) is really more about the genetic consequences of the Milankovitch cycles. However, the sentence does introduce this topic. I suggest replacing (or at least add) Webb & Bartlein (1992).

Added.

Line 75-77 I think perhaps this assumption needs to be unpacked, as it is key to the genomic offset prediction.

We have now added the following reference (Blois et al. 2013, "Space can substitute for time in predicting climate-change effects on biodiversity") here (line 88), which acts as a key reference introducing this concept. With this added reference, and given that such an assumption is common and well-established in the field of climate change prediction, we believe that it should now be sufficiently clear without the need for further elaboration.

Line 117, lines 394-396, Figure 2 and Figure S8. Please clarify the use of EEMS, as it varies throughout the text: reflect population effective diversity rates (q), elevated genetic differentiation. Also the result of EEMS is shown twice (Figure 2A and S8) but the two patterns are different. It is unclear how the analysis differs between these two, given that the method only refer to a single EEMS run. It also seems like there might be some overfitting in Figure S8.

EEMS estimates effective migration and effective diversity rates jointly, i.e. they are two (joint) parameters of the same EEMS model. We have now clarified this by adding this detail in line 433. As detailed in their respective captions, Figure 2A is a map of effective diversity rates while Figure S8 is a map of effective migration rates.

EEMS estimates the two parameters on a dense regular grid (where the edges represent migration and the nodes represent deme diversity). A Bayesian estimation procedure then adjusts the migration rates for all edges so that genetic differences under model closely match that of the observed data. The result is that we can infer discrete and highly resolved patterns in the migration surface (Petkova et al. 2016); i.e., this is not a symptom of overfitting.

Lines 326-329. I am not sure I fully agree with lines 326-329. Results suggest that the populations inhabiting warm elevation environment have shown their capacity to adapt to these warmer condition. While a selective sweep might have reduced adaptive variation, this was likely towards 'sieving' for increasing the frequency of those alleles responsible for a better fitness in warmer environments. Hence directional selection (under warmer climate) should keep on 'sieving' in this same direction. Should we assume that

populations have already 'exhausted' their adaptive potential to warmer environments? Furthermore, the conceptual model shown in figure 1C suggests antagonistic pleiotropy as the main mechanism (a genotype/allele is either adapted or maladapted depending on the environment) but it is also possible that some alleles were adaptive in the past but became invisible to selection (neutral) in modern day conditions (temporal conditional neutrality - de Lafontaine et al. 2018). Such alleles would still bear adaptive potential but simply shifted to a neutral state in contemporary environments. Because they follow the neutral background, they are not expected to drastically decrease in frequency as much as maladapted alleles (when assuming antagonistic pleiotropy) during a sweep. The corollary is that some currently neutral alleles could eventually become visible to selection as climate changes. Hence there would still have plenty of room for future adaptation. I thus disagree with the assumption that low elevation population became more vulnerable, but they certainly are populations worth investigating to assess how marginal populations might bear unique alleles/genotypes/traits conferring fitness advantages that might provide rapid resilience to the species in a warmer climate!

We agree with the reviewer that past sieving in low-elevation populations may have resulted in the increase in frequency of alleles responsible for better fitness in warmer environments, and hence directional selection (under increasingly warm climate) may be expected to keep on sieving in this same direction. However, we show that low-elevation populations are not just depleted for adaptive diversity, but also for genome-wide diversity. Given that the consequences of climate change will not be solely on this axis of (increasing) temperature, but will also include e.g. novel biotic interactions/competition (Steinbauer et al. 2018, Nomoto & Alexander 2021) and shifts in other physical parameters (e.g. precipitation regimes (Anderson & Wadgymar 2020)), we may still thus expect reduced potential for adaptation in the low-elevation populations. We also point to our result that almost all the inferred adaptive variation in the low-elevation populations (i.e. "warm alleles") persists in the high-elevation and refugial populations, i.e. we infer very few unique alleles in the low-elevation populations (new Supplementary Figure S19C). To recognize the nuance in these predictions and address the reviewer's point, we have now added these points to the discussion (lines 360-365) and removed the statement that that low-elevation populations are "the most vulnerable to future climate change".

The comment about temporal conditional neutrality is interesting (indeed, we had initially wanted to include this in Figure 1, but thought it would make the figure overcomplicated). While we do not mention it explicitly in the manuscript (or the concept of antagonistic pleiotropy), we likely account for, and capture some portion of, temporal conditional neutral alleles in our GF models through capturing spatial conditional neutral alleles (e.g. if past or future climate in a certain locale has climate proxies in the present time in a different locale; i.e. a space-to-time analogue). Given that both adaptive and neutral diversity are depleted in low-elevation populations, the source of adaptive alleles (i.e. those exhibiting antagonistic pleiotropy or temporal conditional neutrality) would be depleted too, likely implying loss of adaptive potential to some degree. Whether this loss has any effect on fitness or whether there is still plenty of room for adaptation, however, is uncertain, and we agree that this question would be worth further investigation!

I truly hope these few comments above will at least provide some food for thought!

Best regards,

Guillaume de Lafontaine

Reviewer #2 (Remarks to the Author):

How climate change influences population demography and local adaptation across a species' range has become a huge focus of research. The current study looks at historic changes in climate niche and genomic composition since the LGM to ask whether sorting of climate-adaptive genetic variation occurred during population range shifts away from glacial refugia.

The study represents a sophisticated integration of landscape genomics and climate modeling whose novelty primarily lies with the elegance by which the various analyses come together to address how adaptation proceeded from standing variation during the expansion out of refugia. At some level this result is expected as well as consistent with theory and a wealth of empirical work demonstrating contemporary clines in phenotypes or alleles that are unlikely to have arisen purely from de-novo mutation. However, few studies have produced as large a population-level dataset of whole-genome sequences and related neutral and adaptive components of genomic diversity to the probable location of glacial refugia. Further, none that I'm aware of have shown that genomic composition at climate-associated alleles bear signatures of selection that are proportional to the inferred magnitude of shifting gene-environment relationships (genomic offset). The latter derives from the authors' derivation of a new genomic offset metric that focuses on differences between contemporary and glacial refugial climate, as opposed to most offset studies which focus on current and future climates.

Overall, I found the study very interesting, well written, and meticulously analyzed. I have a few major comments, followed by mostly minor suggestions:

1) I think that the idea behind glacial genomic offset is quite interesting, but I'm not convinced that the approach as implemented captures the signals from both adaptation and IBD/expansion. The formulation calculates pairwise offsets based on refugial climate and soil variables (makes sense), but I don't think that latitude and longitude are good proxies for IBD/expansion (L. 580-583), as they ignore barriers to movement/occupancy and are also inherently correlated with other aspects of the environment. Keeping with the same framework of relating contemporary populations to the predicted refugial site when calculating offsets, you could make use of Moran's Eigenvector Matrices (MEMs) to represent spatial connectivity between your sample populations and the refugium that may better represent expansion history. The edges in the network can be weighted to incorporate knowledge about connectivity (e.g., geographic distance or perhaps other isolating features of topography, habitat suitability, or even better by incorporating routes of expansion from your lineage-specific SDMs). Without a more sophisticated incorporation of the effects of landscape distance and connectivity, I think statements that glacial genomic offset "explicitly integrate the effects of past adaptation, migration, and expansion" (L. 296) are too far-reaching.

We agree with the reviewer that latitude and longitude cannot a priori be assumed to represent good proxies for IBD and expansion. However, in our case, our use of longitude and latitude as proxies was taken following explicit characterization of the species' genetic structure in space. Specifically, both PCA and expansion reconstruction analysis (via the directionality index ψ) describe a geographic cline running east to west. This was supported by the strong correlation of the two main principal components of genetic structure with longitude and latitude ($R > 0.84$, $R^2 > 0.71$; Supplementary Figure S12) and the observation that genetic structure mirrors geography (Figure 3). The use of distance-based Moran's Eigenvector Maps (MEMs) to represent spatial structure (i.e. spatial autocorrelation of samples) as suggested by the reviewer is a good approach when the spatial genetic structure of the sampling is not well-characterised – which we argue is not the case here.

That said, we considered MEMs based on a spatial weighting matrix reflective of the expansion history of the species, as suggested by the reviewer. Specifically, we built a spatial weighting matrix whose edges are weighted by the divergence of expansion paths (from the LGM refugia) between population pairs (lines 598-609; Supplementary Figure S21). This measure of unique expansion routes was calculated over a spatial transition matrix (i.e. resistance or cost surface) defined by the lineage-specific SDM projection, to account for the heterogeneous landscape of the Alps (Supplementary Figure S21A). A single, positive MEM eigenvector (Supplementary Figure S21B) explains almost the entirety of the structure/variation here, and expectedly, recapitulates our PCA and expansion (ψ) plots (Supplementary Figure S21 C; Figure 3A). Regression of this distance metric with F_{ST} (isolation-by-distance analysis) shows it to explain population differentiation well (Pearson's $r = 0.681$, p -value $< 1 \times 10^{-4}$), though again notably, virtually identical to that (IBD) based on great-circle distance ($r = 0.687$, p -value $< 1 \times 10^{-4}$) (Supplementary Table S3). This implies that this highly parametrized MEM eigenvector explains IBD/genetic structure approximately as well as longitude-latitude; i.e. this MEM eigenvector is a simple representation of the two geographic coordinates. As such, this MEM eigenvector is also similarly correlated with other aspects of the environment.

An important limitation of using MEMs in our study is that while it allows us to (partially) control for the spatial structure (via acting as a spatial co-variate), it is not straightforward to use it to model the effect of geographic distance in projected space and time (i.e. where we have no MEM layers). The inclusion (rather than removal) of distance effects is something we explicitly desire in our glacial genomic offset, for we envision it to reflect both adaptive effects (due to environment) and distance effects (due to IBD and expansion). This is in contrast to most applications of MEMs where distance effects are undesired and intentionally excluded (e.g. if looking purely at adaptation such as in the classical genomic offset). There is no straightforward way to generate glacial genomic offset predictions with MEMs, barring (potentially crude) interpolation which can be highly artifactual especially when sampling is uneven and the landscape heterogeneous. We nevertheless explored this method via an inverse path distance weighting approach that interpolated MEM values across the landscape respecting the cost surface of the landscape (lines 643-647). The resultant glacial genomic offset calculated with this interpolated MEM layer is qualitatively similar to our old prediction (calculated with longitude and latitude as co-variables) but at the cost of complexity (large degrees of parametrization), and notably with various artifacts that resulted from the interpolation of few and uneven points (Supplementary Figure S21D). We have added these new results to the manuscript (line 598-609, 643-647; Supplementary Figure S21) but retain our original glacial genomic offset predictions in the main figure and results, given the limitations and caveats of MEMs mentioned above.

2) In projecting offsets back to the LGM, it seems that you're also extrapolating beyond the range of climate values represented by contemporary populations used to train the GF model. This climate novelty is apparent in Fig 4B. It's not known how this extrapolation is likely to bias offset estimates, and this point should be acknowledged and discussed for how it may impact your inference.

While there are indeed combinations of environment (i.e. climate novelty) unique to the LGM (Figure 4B), the portion of environmental space represented in the LGM refugia (now shown in the revised inset of Figure 5A) is a subset of that in the present-day (used to train the data). Because we rely only on the environmental space represented in the LGM refugia (and not the whole Alpine region) to calculate the glacial genomic offset (lines 630-638, Supplementary Figure S20), we do not rely on any extrapolation (to novel climate), meaning our offset estimates should be robust in this context.

Introduction: In my opinion, the second paragraph sets up a bit of a false dichotomy, specifically where the argument is developed that post-Quaternary migration with no adaptation was/is the assumption, and that only recently has this paradigm been challenged. While Davis and Shaw (2001) was a landmark paper that

made the assertion that adaptation should be considered as part of the post-glacial process, that paper is now >20 years old, and a huge literature now focuses on adaptation during range expansions. I would suggest toning this down – still make the point, but don't force this as a recent paradigm shift. Here also I think the concept of selection on standing genetic variation (SGV) could be usefully introduced to complement the “sieving” analogy.

We have now toned this down in the introduction (line 69). We have also added detail of selection on standing genetic variation in the context of “sieving” in the introduction (lines 73-74) and to Figure 1 (new panels C and D).

Minor comments:

L 88: “leverage on the expectation” – awkward phrasing

We have now rephrased this sentence (lines 98-101)

L. 128: I think most readers will interpret the phrase “strong reproductive isolation” to mean the presence of intrinsic (genetic) barriers to mating. But the lack of admixture you've shown here could be due to several different causes. Perhaps better to say, “a prolonged history of isolation”.

We have now changed this according to the reviewer's suggestion (line 147)

L. 185-186: Isn't it circular to conclude that your lineage-specific SDMs generate distributions highly congruent with the genetic structure? The occurrences feeding into these SDMs are being defined by the genetic structure of these lineages.

We should first clarify that the occurrence data feeding into the SDMs are separate from the genetic samples (lines 520-526), and were delineated by taxonomic denomination and geography rather than directly by genetic sequencing. While the reviewer is correct that genetic lineage assignment of geographic occurrences was informed by the inferred genetic structure, we argue that this is not entirely circular. To illustrate this, we point to our SDMs based on lineage-specific occurrence data that lead to the SDM projections (shown in Supplementary Figure S14 right column), which are very different than the pattern describing genetic structure (Figure 2A; Supplementary Figures S4 & S5). Only once we incorporate refugia, dispersal and competition do we recapitulate the observed current distribution (Figure 4A; thus implying that these processes were key in defining their current distribution).

L. 188: It seems to me that competitive exclusion between your lineages is more of an assumption you're making (out of convenience) in the modeling than a result that has statistical support. If you do want to show the latter, it would seem to require some sort of model selection process among variables that do/don't include that assumption.

Our inclusion of competitive exclusion derived from the observation (both personal and from species occurrence records) that the different lineages are not observed to grow in sympatry – contact zones are often highly discrete even without evident difference in environment across contact zones. We used this prior knowledge, rather than statistical model selection, to constrain the model as such.

L. 227: It's a misinterpretation to refer to “the adaptive genotype” when interpreting the visualization of genomic composition in Fig 5. These maps represent ordinations of population-level allele frequencies from across the genome. See also L. 234-235 (“predicted genotype”).

Here, we used the term genotype in the broad sense, i.e. referring to the genetic makeup of a population, rather than in the narrow sense i.e. referring to alleles or gene variants. We used this term over the more specific term “adaptive genomic composition” because we found the latter phrasing unwieldy if used too frequently in the text. To avoid potential misinterpretation, however, we have followed the reviewer’s suggestion and have now changed “adaptive genotype” to “adaptive genetic composition” in the caption of Figure 5 and in line 257 (previous lines 234-235).

L. 246: I quite like this set of analyses relating glacial genomic offset to π and the frequency of derived alleles, but I wouldn’t consider this “validation”. Rather, I think these analyses corroborate the offsets, but do not independently validate them.

Here, we test predictions (glacial genomic offsets based on modelled historical processes) against observation (contemporary population genetic data). In such a case, we believe “validate” is an appropriate term to use, as is “corroborate”. We agree, however, that these sets of analyses are not independent, but we do not state that this is the case in the manuscript.

L. 263-264 and Fig 6B: This is a nice result (higher nucleotide diversity at env-associated loci) but lacks statistical assessment. Can you include a Mann-Whitney test here between sets of loci? I would also temper the conclusion about highly diverged haplotypes (that hasn’t specifically been shown). Lastly, it would be helpful to see the distributions of π values (e.g., in violin plots) rather than the point estimates in Fig 6B.

We have now added a Mann-Whitney test of significance between nucleotide diversity at environmentally-associated loci versus genome-wide (lines 291-292). Regarding our reference to highly diverged haplotypes, we phrased it as a suggestion (“suggestive of”; line 292) rather than a conclusion. We have actually tested the hypothesis of highly diverged haplotypes in some detail and will report detailed results in a separate manuscript (Fior et al.; see also response to next question). Regarding having Figure 6B show π distributions rather than point estimates, we considered this, however, found that the resulting figure would look too dense, as the range of π spans several orders of magnitude and would overlap between π_{GF} and π_{GW} . That said, we agree that showing the distribution of π is useful and we have now added these for an example pair in the supplements (Supplementary Figure S22) for GW, GF and the top 1000 GF SNPs. The distributions for the other population pairs appear qualitatively identical.

L. 356: Will the Fior et al. manuscript that describes the reference assembly used here for mapping be available prior to publication of the current paper? If not, consider making the genome assembly itself publicly available in a curated repository.

This Fior et al. manuscript will likely not be available prior to the planned publication of this paper, though it may appear as a pre-print by that time. That said, we have made the genome assembly public available via depositing in a public repository (<https://doi.org/10.5061/dryad.x0k6djhnq> (permanent link, will be made public at time of manuscript publication); <https://doi.org/10.5061/dryad.x0k6djhnq> (temporary link; currently made public)), as indicated in the Data availability section.

L. 364: Here, and throughout, “principle” should be “principal” when referring to PCA.

This has now been corrected.

Fig 1B-C: I’m not sure about the patterned fill on these distributions...I think simple solid colors with partial transparency would be less distracting.

Distribution curves in this figure are now given solid, monochrome colours (shades of grey), to be less distracting.

Supplemental:

L. 88-89: Did you hard-call genotypes prior to phasing?

No, the inputs to Beagle phasing and imputation were ANGSD genotype likelihood files. This detail has been added to line 96 in the Supplements.

Fig S16: What does the shading represent in this map? DEM?

Indeed, the shading in Figure S16 refers to elevation (DEM). We have now added this detail to the figure caption.

Reviewer #3 (Remarks to the Author):

Luqman et al. use an impressive genomic database (whole genome resequencing in over 1200 individual & 115 populations) to analyze the evolutionary history of *Dianthus sylvestris* populations. With this data, the authors show that both adaptive and demographic processes have influenced the current distribution and shaped the genetic diversity of this species. Using clustering algorithms, species distribution models and demographic inference, the authors show that three lineages diverged during the PGIP, and remain well differentiated after having independent histories (three distinct refugia & low gene flow between groups). Second, the authors show that the Alpine populations had a recent expansion from their refuge, and that most populations grow in conditions similar to their respective refuge. They also show that the rest of the populations grow at lower altitudes and different conditions, suggesting they might have adapted to those conditions (by the sieving of alleles). Third, they find that populations growing at lower altitudes or at expansion routes have a higher “glacial genomic offset”, suggesting that IBD and adaptation have shaped the genetic composition of populations. Importantly, the authors find that higher glacial genomic offsets are consistent with increased selective sweeps in low altitude populations, supporting their hypothesis. The authors conclude that the distribution and genetic diversity of *Dianthus sylvestris* is explained by a combination of migration and “sieving” of adaptive variation.

I think this manuscript is very interesting, and tests a very important question in evolutionary biology: How do populations respond to climate change? In short, their results suggest that populations can respond rapidly by migrating to suitable environments and simultaneously adapting to locations where they can tolerate new environments. I also think that the manuscript is very well written and that the methods they use are very complete and robust (although I have a few suggestions, see below). For example, I really like that the authors combined adaptive and neutral genetic diversity with environmental data in their analyses. I also think that the results are very robust, since the authors considered plenty of factors that could generate false positives or biases in their analyses (i.e., consider number of individuals sampled in clustering analyses; comparing admixture and chromosome painting tests; consider geographic patterns of isolation and colonization; using different algorithms to perform species distribution models; using different climatic circulation models; testing consistency between methods; comparing neutral vs climate related genes; corroborating glacial genomic offset with SFS deviation tests; contrasting altitude patterns vs geographic patterns; discussing the limitations of analyzing glacial genomic offsets using SDMs and assuming niche conservatism, etc.). Finally, I think they discuss critically many of the limitations in their analyses using important literature.

In sum, I think this is a very interesting paper, very well written and methodologically very robust. I think this paper is suited for Nature communications since it will be of interest for a very broad range of scientist.

Below, I address a few simple comments that I think would improve the paper.

1) Sieving hypothesis: I really like this hypothesis, and I think it is very relevant to combine adaptive and demographic data to understand evolution. In lines 61-63 & 69-71, the authors explain the hypothesis. However, I do not think that the hypothesis is explained clearly which affects the punch of the introduction. Also, I think that Figure 1 is not very intuitive. I am not a native English speaker (although I am fluent), and I did not know the word “Sieve”. Until I searched for the definition, I was able to finally understand the hypothesis well. I think that a clearer explanation of the hypothesis would benefit the flow of the manuscript. Perhaps explaining step by step the sieving mechanism (migration first and selection second acting on adaptive alleles) might be helpful. I also think that Figure 1 could be simplified by adding a graphic representation of the mechanism of sieving of a population moving across a landscape (a simple schematic

image). I also think that the shapes in the density plots add unnecessary complexity and are not esthetic. I would change to simple colors (I like the palette in Figure 2A) or shapes of grey.

Following the reviewer's recommendation, we have now added a conceptual schematic of the sieving process in Figure 1 (panels D and E), to aid in interpretation. We have also elaborated the sieving process in the introduction (lines 73-74).

2) Following this idea, I was puzzled that the authors mention the importance of allele sieving, but I did not find any result that explicitly explores the sieving of those alleles. The authors use the GF loci to compare environmental related SNPs vs genome-wide SNPs. I get that the authors consider that analyzing the entire contribution of environmental related SNPs can show genome wide patterns of adaptation. I also get that there are low number of suitable pairs (1606-607). However, I think it would be very interesting to perform more explicit analyses. For instance, the authors could do a genome environmental association analysis and detect selective sweeps that segregate in populations growing in alpine and low-altitude populations. In figure 3C the authors show that altitude does not correlate with geography suggesting that there are multiple altitude gradients and perhaps parallel evolution. The authors also find signals of selective sweeps in low altitude populations (Figure 6). I think it would be very interesting and valuable to test if there are alleles that in parallel are selected in those populations. It might be as simple as to test for environmental related GF alleles, if they correlate with altitude but not geography. Perhaps a NJ tree with adaptive alleles would group lowland populations and this could be contrasted with the genome-wide effects. If shared alleles occur in populations at different locations, it could indicate that adaptation from standing genetic variation is occurring.

We have now added two new sets of analyses that more explicitly highlight the sieving process in our system, as requested by the reviewer. First, we now show the relationship between low- and high-elevation individuals based on the top 1000 environmentally-associated SNPs via genetic distance tree and PCA (Supplementary Figure S19). As expected, the tree and PCA results recapitulates our GF results that low-elevation populations share (recruited) a common set of adaptive alleles (i.e. suggestive of parallel adaptation), which is divergent from that of the high-elevation populations. Second, we assessed the presence and absence of adaptive alleles (top 1000 environmentally-associated SNPs) among high-elevation, low-elevation and refugial-proxy populations, which we present (via a Venn diagram) in Supplementary Figure S19C. Our results show that high- and especially low-elevation populations recruited subsets of available standing genetic variation available in the refugia, as expected under the proposed sieving process.

We have already performed genome environmental association analysis (gradient forest is explicitly a GEA method) and explored genome-wide signals of selective sweeps within and between high- and low-elevation populations (based on population genetic analyses of selective sweeps; Figure 6). Similarly, we have already inferred patterns of parallel adaptation (e.g. among low-elevation populations) as previously described in lines 243-248 and visualized in Figure 5A (right panel). Specifically, shared (and contrasting) composition of adaptive variation are quantified and visualized as similar (or divergent) colours. This allows for the explicit comparison of low- vs. high-elevation populations, as suggested by the reviewer; e.g. showing that low-elevation populations are shown to exhibit similar composition (frequencies) of adaptive alleles but contrasting to that of high-elevation populations. This is in contrast to the neutral or genome-wide pattern of structure (Figure 3).

With the inclusion of the new analyses, we believe our manuscript now provides sufficient evidence for the spatio-temporal sieving hypothesis for adaptive genetic variation in our system. We agree that more can be done to dissect the process of adaptation, e.g. by looking in-depth at the selection signature and perhaps even function at particular candidate adaptive loci rather than genome-wide signals of adaptation (as performed here). Such work is in process and will be presented in separate manuscripts

(e.g. the Fior et al. manuscript referenced in the text and mentioned in reply to Reviewer 2's comments).

3) Glacial genetic offsets: I think these methods and results are really interesting and smart. These results are important because they suggest that genetic offsets are powerful approximations to detect how populations will adapt to changing environments. In general, most studies that have used genetic offsets have the limitation that corroborating the results is complex. Only a few studies have included experimental validations. By analyzing patterns from past to present, the authors are able to test their results based on the validation of genetic diversity patterns (perturbations of SFS analyses).

Having said that, I think that the results could be presented better. First, I think that for people who are not familiarized with genetic offsets, it would be convenient to present a schematic figure of how glacial genetic offset from refuge to current locations were estimated (the methods are very well written, but a schematic representation might help). Glacial genetic offset is a complex concept and this is a journal for a broad readership. Second, I agree that the results indicate that the current distribution of alpine populations has a similar projected genetic composition than the refuge in the past. However, there is no statistical analysis or metric that truly shows this. This is a problem, because the authors also support their sieving hypothesis by saying that low altitude locations grow at different climatic distributions. However, since the LGM refuge is so small it is not easy to analyze the different colors that are present in the refuge. For example, it is not clear if the "green-bluesish green" distribution did not exist in the past. I think that a good way to compare genetic compositions between present and past distributions (black polygons), would be to replicate the byplots in Figure 5A and only show the genetic composition distribution for locations within the polygons. This way it would be possible to see if in the byplot, the present alpine distribution overlaps with the refuge distribution; and to see if lowland populations also overlap with some of the past distribution. Finally, the authors could also differentiate between high and low altitude populations and use either centroids or distribution of PC scores to compare if the genetic compositions are significantly differentiated from the refuge genetic composition.

We have now added schematic representation of how we calculated the glacial genomic offset (Supplementary Figure S20), to aid in interpretation. Following the reviewer's suggestion, we have also now added contours to the biplot of Figure 5A, encapsulating the environmental (biological) space occupied by the LGM (refugia) and present-day ranges. This shows which range of "colours" overlap or are unique to the two time points, thus representing how the temporal expansion of the adaptive genetic composition of the species is mapped on the spatial distributions. For a quantitative comparison of the difference between LGM refugial populations and contemporary high and low elevation populations, this is precisely what the glacial genomic offset does. From Figure 5 and lines 258-265, we show that the populations inhabiting the low-elevation valleys are among the most diverged in adaptive genetic composition (offset > 0.01; reds) compared to the refugial populations during the LGM, while alpine areas are generally closer (offset < 0.01; blues; contingent on the expansion distance from the refugia). The value of the offset reflects, in some manner, the ensemble allele frequency shift of adaptive alleles (since it is modelled from the ensemble allele frequency responses of many SNPs), and can hence be interpreted as the amount of evolutionary change between the source (refugial) population and their contemporary descendants. The offset results are presented as a continuum in space (similar to most predictive climate models, e.g. SDM, classic genomic offset, etc.), rather than discretely for the two categories (high and low), to retain maximum information and avoid lossy discretization (binning of GF predictions). While we can project sampled high and low elevation populations in the biplot (Supplementary Figure S18), there are too few samples (n = 10 and 26 for low- and high-elevation populations, respectively) to perform tests of significance.

4) Why didn't the authors perform the glacial genomic offset analyses for the other genetic groups? This could go in supporting information. Is it because their potential distributions have been more conserved?

We did not perform glacial genomic offset analyses for the other two lineages because this approach requires a detailed characterisation of expansion history, which could not be achieved with the limited number of populations (comprising sufficient number of individuals for accurate allele frequency estimation) available for these groups (Table S1). For the Alpine lineage, we used the directionality statistic ψ , which (like gradient forest) relies on accurate estimates of allele frequencies and dense coverage of populations in space. For comparison, we hereby provide a sampling overview of the three lineages:

- *Alpine: 930 individuals, 67 populations, 43 populations with sufficient individuals (here taken to be 14) per population for accurate allele frequency estimation*
- *Apennine: 160 individuals, 22 populations, 4 populations with sufficient individuals per population for accurate allele frequency estimation*
- *Balkan: 171 individuals, 26 populations, 4 populations with sufficient individuals per population for accurate allele frequency estimation*

5) The authors use the names of geographic locations to point some results (i.e., Monte Baldo, Dolomites, etc.), however, these names are not present in the maps. For people that are not European, or not familiarized with these locations, it is a bit hard to follow. I think it would be convenient to add some of these names in the Figures or see if it is possible to only use North, South, East, West.

We have now added these place names to Figure 3. Additional place names are indicated in Figure 2.

6) I think it would be convenient to make a supporting figure that shows the overlaps between the LGM and present Sdms of figure 4A. I would make one for each genetic group. This would help visualize the possible refugia. For the Alpine group, this is clear, but for the other two not so much.

Figure 4A already shows the LGM distribution (i.e. refugia) for each genetic lineage. I.e. there is no need to find the overlap of the present and LGM projections for this. To avoid potential misinterpretation, we have now added this detail (point of clarification) to the figure's caption.

7) I would like to see a short description or justification of why *Dianthus sylvestris* was used as a model.

We have now added this (lines 98-101).

8) How important is standing genetic variation for this hypothesis? I think this should be discussed more.

Standing genetic variation is an important part of the hypothesis, as it acts as a source of adaptive genetic variation on which "sieving" acts. We have now added a sentence mentioning this (line 73-74) as well as conceptual illustration that visualizes this (Figure 1D).

9) In figure 6 B & C, it is not very clear the distinction between π_{GF} and π_{GW} . At the right two square brackets indicate this for Figure B and C but it is not easy to follow. Instead, I would change to colors indicating high and low altitude populations, and triangles and circles to indicate π_{GF} and π_{GW} , respectively.

We have now revised this figure, such that colours now denote high and low elevation populations, and triangles and circles now denote π_{GF} and π_{GW} , respectively, as suggested by the reviewer. While revising this figure, we also noted a miscalculation in our estimates of π_{GF} (off by an order of magnitude). We have now corrected this, and replotted π in log scale, to accommodate the large range in π values.

Reviewers' Comments:

Reviewer #1:

Remarks to the Author:

All my concerns were successfully addressed in the revised version. Figure 1 is much clearer now. I only have two minor edits.

Line 584 : For the sake of consistency throughout the article, replace sorting by sieving
Title of the supplementary information should match the title of the article.

Reviewer #2:

Remarks to the Author:

I reviewed the first version of this ms, and find the revisions made by the authors in this updated version to be very thorough in response to feedback from myself and the other reviewers. I find this study presents an extremely well analyzed and interesting case study of migration and adaptation during expansion from glacial refugia, and I recommend its acceptance for publication.

Reviewer #3:

Remarks to the Author:

I have re-read this manuscript and I believe that the authors have addressed all my concerns and have responded or addressed the concerns of the other reviewers. I think figure 1 is very clear now.

I do not have further comments!

As I mentioned in my previous review, I think that this manuscript is really interesting, very well done and that it will contribute to our understanding of how populations adapt to climate change. I am positive that this manuscript will be a great fit in Nature Communications and I look forward reading it in the final format.

Best wishes

Jonás Aguirre

RESPONSE TO REVIEWERS' COMMENTS

Reviewer #1 (Remarks to the Author):

All my concerns were successfully addressed in the revised version. Figure 1 is much clearer now. I only have two minor edits.

Line 584 : For the sake of consistency throughout the article, replace sorting by sieving
Title of the supplementary information should match the title of the article.

We have replaced sorting with sieving on line 598/623 (Word/PDF version; previously line 584), and revised the title of the supplementary information to match that of the main article. Thank you for the feedback.

Reviewer #2 (Remarks to the Author):

I reviewed the first version of this ms, and find the revisions made by the authors in this updated version to be very thorough in response to feedback from myself and the other reviewers. I find this study presents an extremely well analyzed and interesting case study of migration and adaptation during expansion from glacial refugia, and I recommend its acceptance for publication.

Thank you for the feedback and recommendation.

Reviewer #3 (Remarks to the Author):

I have re-read this manuscript and I believe that the authors have addressed all my concerns and have responded or addressed the concerns of the other reviewers. I think figure 1 is very clear now.

I do not have further comments!

As I mentioned in my previous review, I think that this manuscript is really interesting, very well done and that it will contribute to our understanding of how populations adapt to climate change. I am positive that this manuscript will be a great fit in Nature Communications and I look forward reading it in the final format.

Thank you for the feedback and recommendation.